# Evolving affinity between Coulombic reversibility and hysteretic phase transformations in nano-structured silicon-based lithium-ion batteries

K. Ogata et al.#

Nano-structured silicon is an attractive alternative anode material to conventional graphite in lithium-ion batteries. However, the anode designs with higher silicon concentrations remain to be commercialized despite recent remarkable progress. One of the most critical issues is the fundamental understanding of the lithium–silicon Coulombic efficiency. Particularly, this is the key to resolve subtle yet accumulatively significant alterations of Coulombic efficiency by various paths of lithium–silicon processes over cycles. Here, we provide quantitative and qualitative insight into how the irreversible behaviors are altered by the processes under amorphous volume changes and hysteretic amorphous–crystalline phase transformations. Repeated latter transformations over cycles, typically featured as a degradation factor, can govern the reversibility behaviors, improving the irreversibility and eventually minimizing cumulative irreversible lithium consumption. This is clearly different from repeated amorphous volume changes with different lithiation depths. The mechanism behind the correlations is elucidated by electrochemical and structural probing.

Correspondence and requests for materials should be addressed to K.O. (email: k.ogata@samsung.com) or to S.J. (email: seongho.jeon@samsung.com) or to S.H. (email: sungsoo1209.han@samsung.com). #A full list of authors and their affliations appears at the end of the paper.

Si is an attractive alternative to the commonly used graphite (Gr) as the negative electrode in the Li-ion battery (LIB), owing to its significantly high specific capacity (~3579 mAh $g^{-1}$ at room temperature, assuming $Li_{3.75}Si$)[1]. However, the high capacity is associated with huge volume changes (~270–300%)[1], which cause capacity loss and prolonged irreversible reactions. To develop the large-scale application of Si, recent studies have examined some elaborately engineered Si composites[2–15] that can reasonably accommodate the volume changes and retain the capacity over hundreds or thousands of cycles[2,3,11–13,15,16]. Further, new in situ and ex situ analytical methods have also helped to understand the underlying mechanisms[17–30]. These studies show that crystalline-Si (c-Si) is converted into amorphous-$Li_xSi$ (a-$Li_xSi$) phases during the first lithiation, which involves large asymmetric volume changes[22] owing to different Li reaction rate constants at different c-Si facets[31]. Upon lithiation, a-$Li_xSi$ transforms inhomogeneously into metastable crystalline-$Li_{3.75}Si$ (c-$Li_{3.75}Si$) at low voltages (<70 mV vs. Li)[21,23–27,30,32], and overlithiated phases such as c-$Li_{3.75+\delta}Si$[23,25] at room temperature, which are associated with a large overpotential on delithiation (430–450 mV)[26,30], different Si/passivation-layer interface formation[33], and extra capacity loss[7,26,30]. Li–Si processes on (de) lithiation can be either asymmetric (i.e., with c-$Li_{3.75(+\delta)}Si$), or symmetric (without it), because of the complex Li–Si energetics[25].

Despite all these insights, anodes with higher Si concentrations have not emerged on the market. One of most critical bottlenecks is capacity loss via prolonged irreversible Li consumption in the Li–Si processes, which is often quantified by Coulombic efficiency (CE, the delithiation/lithiation capacity ratio)[34–36]. This is because in practical full cells the supply of Li atom is limited by the cathode loading, unlike the case of unlimited supply in Li-metal-countered half-cells. Also, while state-of-the-art commercial Si/C composites can somewhat manage the volume change and capacity decay to achieve longer cycles, these composites cannot sustain CE at a higher level over longer-term cycles. This is particularly serious when the Si concentrations in the composite are higher. CE is strongly associated with the formation of by-products (i.e., solid electrolyte interphase; SEI)[33,37–39] at the Si–electrolyte interface, and/or Li trapping in Si owing to the unique volume changes on (de)lithiation. Hence, one intuitive strategy to achieve higher CE is to limit excessive electrolyte invasion into Si interface, by forming protective shells/coating around Si[2,16,40]. Nevertheless, electrolyte can still invade due to the transport of Li ions (coupled with organic components) and/ or gradual deformation of the composites upon iterative volume changes even with engineered internal pores. Hence, scenarios with Si exposed to electrolytes should be considered for understanding the CE fundamentals.

One of the most basic is to understand the evolving CE alterations by different Li–Si reaction paths over longer cycles, when the electrode is fully exposed to representative electrolytes. More specifically, the key is to quantitatively separate contributions to CE alteration from the incremental volume change in the amorphous Li–Si and that due to amorphous–crystalline (a–c)-Li–Si phase transformations. However, such studies are surprisingly scarce. Importantly, such information requires a few prerequisites that are unfortunately not considered in previous lines of work. Firstly, the error in CE determination due to instrumental precision and electrode reproducibility should be well-defined and sufficiently suppressed, in order to examine potentially small CE alterations. Secondly, it is necessary to reference the experimental Si reversible capacity in the first cycle to the theoretical value, so that the Li–Si lithiation depth or namely depth of discharge (DOD) in the half-cells in the following cycles can be numerically controlled by capacities. Further, the electrodes need to be designed to ensure that the a–c transformation abruptly occurs near DOD100% even at higher current rates. Without these considerations, it is very difficult to resolve subtle CE alterations over longer cycles, or quantitatively separate the influence by mere amorphous Li–Si volume changes from that by the a–c phase transformations. There is a common misunderstanding that exploring the a–c transformation is not important, because the practical state of charge (SOC) for the anode in full cell systems is usually less than 100%, i.e., with an average $x < 3.75$ in $Li_xSi$. However, this is not really the case at realistic current densities, because the Li–Si processes involve strong variations in Li concentration across the electrode and overpotentials under kinetic cycling conditions (see Methods section under "Baseline active materials"). Hence, revealing such CE alterations by using a deeper range of DOD has significant importance.

In this study we quantitatively and qualitatively separate the CE alteration by incremental amorphous Li–Si volume changes from that by the a–c transformations, by precisely controlling DOD% in a series of Li-rich Li–Si phases. The electrodes are designed to satisfy the above-mentioned prerequisites. The electrochemical probing of CE alterations is further combined with various atomic-scale methods such as ex situ X-ray absorption fine structure (XAFS), ex situ magic-angle-spinning solid-state nuclear magnetic resonance (MAS ss-NMR), density functional theory (DFT) calculation, ex situ X-ray diffraction (XRD), ex situ (scanning) transmission electron microscopy (TEM/STEM), and ex situ X-ray photoelectron spectroscopy (XPS). From these complementary approaches, we show that the cumulative c-$Li_{3.75(+\delta)}Si$ formation/decomposition over cycles and consequent changes in the structural/interfacial characteristics are key for governing the CE behaviors. Thus, for the first time we highlight how the a–c transformations, typically featured as a degradation factors, can benefit the practical full cell systems.

## Results

**Perspective of experiments**. The overall experimental scheme in this study is illustrated in Fig. 1. Figure 1a shows different Li–Si electrochemical reaction pathways on (de)lithiation over cycles, which are quantitatively and qualitatively controlled by the DOD (70–100%). Subsequently, various electrochemical outputs for different DOD controls are analyzed as shown in Fig. 1b. The electrodes are mainly cycled at 1 C under given DOD controls. Yet on every 20th cycle, all electrodes are slowly cycled under DOD100% regardless of the DOD value used in previous cycles, in order to capture the structural characteristics for the full potential range (Fig. 1c). These cycles are the probing points that are discussed in the following sections. A list of electrochemical/ structural probing is shown in Supplementary Table 1. In the following sections, we firstly explain the design principles of the active materials and electrodes (Fig. 2), followed by discussing evolution of the electrochemical Li–Si processes (Fig. 3) and Coulombic reversibility (Fig. 4) under different DOD values over cycles. TEM (Figs. 5 and 6), XRD (Fig. 6), MAS ss-NMR (Fig. 7), and XAFS (Fig. 8) methods are further used to reveal the mechanism associated with the reversibility. Finally, all obtained mechanistic results associated with the irreversible behaviors are schematically summarized in Fig. 9. The detailed electrochemical and mechanistic findings are also summarized in Supplementary Table 2 and 3, respectively.

**Active materials and electrodes**. Two active materials (named type-A and -B) are fabricated via a spray-drying method as shown in Figs. 2a–d, to confirm the consistency of electrochemical outputs for electrodes with different Si concentrations. The active materials consist of commercially available polycrystalline Si

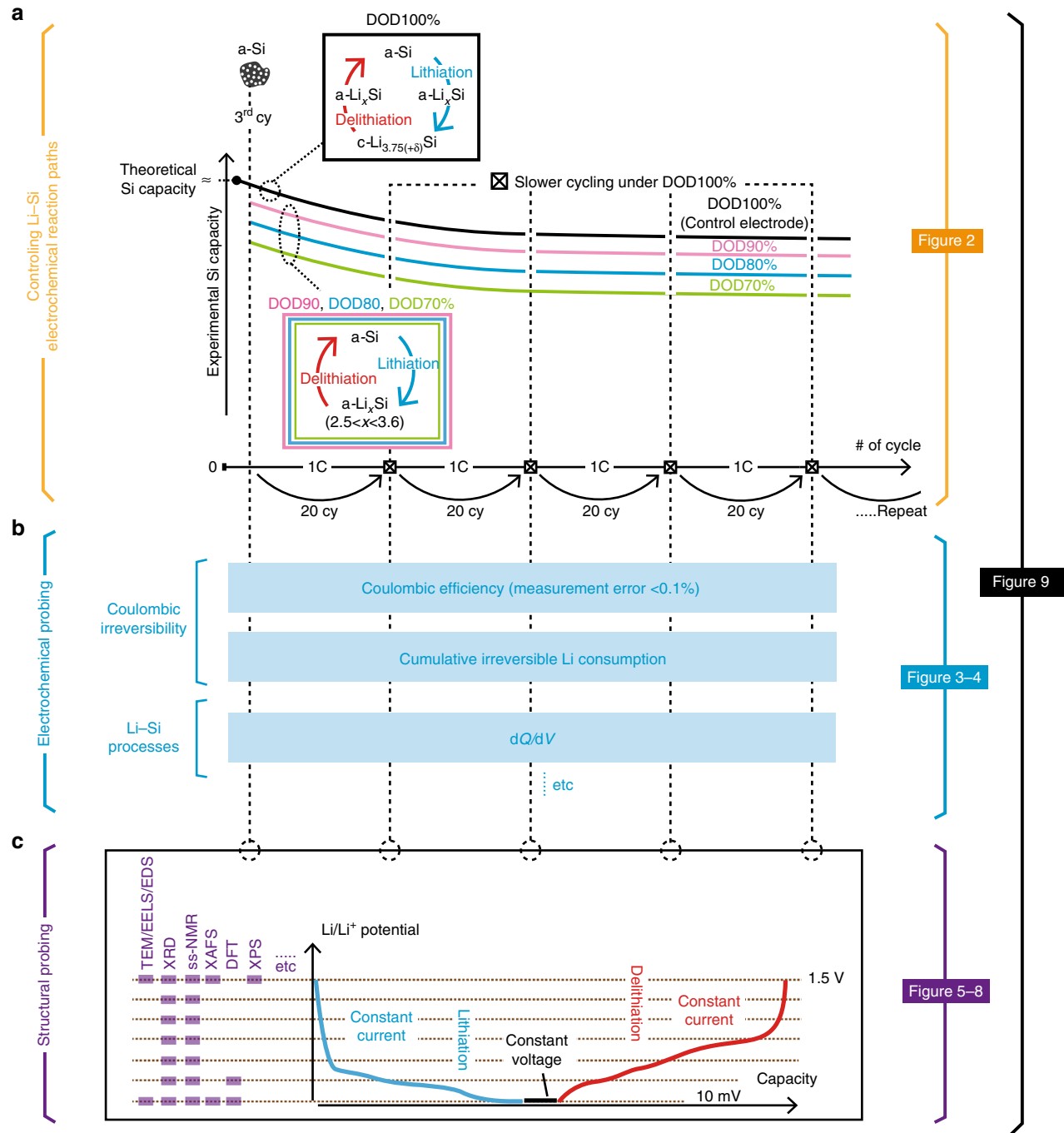

**Fig. 1** Schematics of experimental scheme. The overall experimental scheme is illustrated in (**a**–**c**). **a** Li–Si electrochemical reaction paths are controlled by capacity-cutoff depth of discharge (DOD)%. After polycrystalline Si nanoparticles (pc-SiNP) are fully amorphized in the first two cycles, DOD% is controlled from the third cycle on to 70, 80, 90, or 100% (indicated by green, blue, pink, and black solid lines, respectively). As shown in colored rectangular boxes, under the DOD100% protocol, the system displays an amorphous–crystalline Li–Si phase transformation, while DOD70–90% displays amorphous–amorphous volume changes. **b** Electrochemical outputs to be probed under different DOD controls, such as Coulombic efficiency (CE), and cumulative irreversible Li consumption, and dQ/dV profiles. The capacity of electrodes is carefully designed such that the first reversible capacity under DOD100% agrees with the Si theoretical capacity to ~99% accuracy (see Methods under "Baseline active materials"). **c** Various analyses used for structural probing, which is conducted every 20 cycles at slower current rates under DOD100%, in order to capture the structural characteristics for the full potential range for different DOD controls (see Methods under "Electrode fabrication and cycling conditions"). Figures corresponding to (**a**–**c**) in the following section are labeled on the right side

nanoparticles (pc-SiNP) and multiwall carbon nanotubes (MWCNT) with or without flake-type graphite (Gr; for details see Supplementary Table 4 and Methods under "Baseline active materials"). The electrodes are assembled in 2032-type Li-metal-countered coin half-cells, and they are cycled under constant current constant voltage (CCCV) mode on lithiation, and constant current (CC) mode on delithiation (Methods under "Electrode fabrication and cycling conditions"). The active materials are designed to form a porous structure for better wettability (Figs. 2c, d) and ensure the accessibility of Li ions to Si surface.

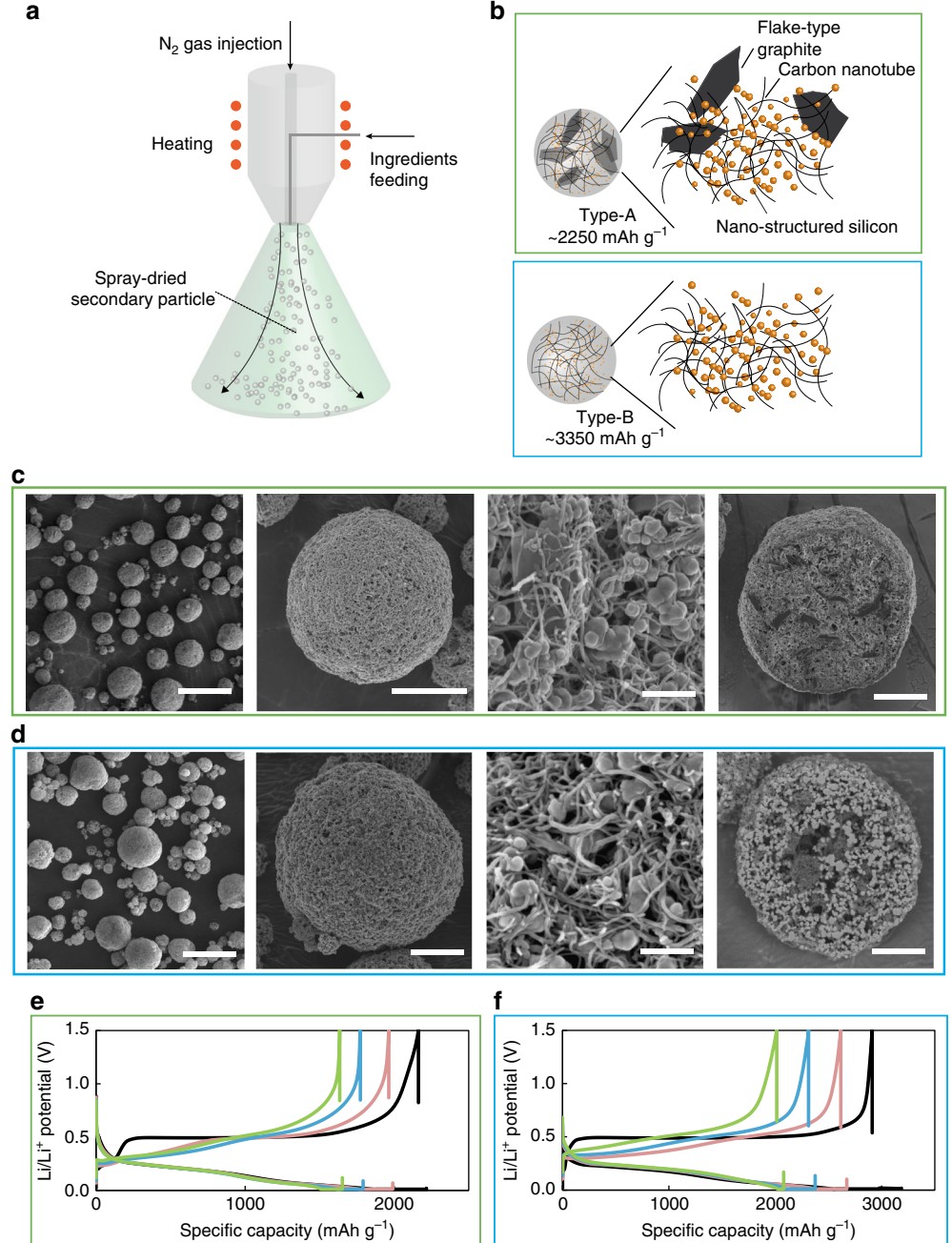

**Fig. 2** Preparation of active materials with different Si concentrations. Schematic of (**a**) the spray-drying process for secondary particle fabrication and (**b**) spray-dried type-A and -B secondary particles (with the first reversible capacity being 2250 and 3350 mAh g$^{-1}$, respectively) as active materials. Particles with two different Si concentrations are used in this study to confirm consistency of the electrochemical outputs with different Si concentrations. For more detailed material properties for these particles see Supplementary Table 4. The schematics for type-A and -B are surrounded by green and blue empty rectangles in (**b**), respectively. SEM/BSE images of (**c**) type-A and (**d**) type-B secondary particles. The three images on the left in (**c**, **d**): planar views by SEM with different magnifications, on the right in (**c**, **d**): FIB cross-sectional views by BSE. The scale bars in the images (from left to right) in (**c**, **d**) are 20 μm, 5 μm, 200 nm, and 5 μm, respectively. (**e**, **f**) Li/Li$^+$ potential as a function of specific capacity on the 10$^{th}$ cycle for different capacity-cutoff depth of discharge (DOD) percentages for (**e**) type-A and (**f**) type-B electrodes. The black, pink, blue, and green profiles correspond to cycling under DOD100%, 90%, 80%, and 70%, respectively

Also, the initial reversible capacities under DOD100% are adjusted to the theoretical capacity of Si to ~99% accuracy in the 1$^{st}$ cycle (see Methods under "Baseline active materials"). The amorphization process of polycrystalline SiNP over the first two cycles is followed by a capacity-controlled DOD regime at 1 C, in which DOD70–90% capacity profiles are programmed, referring to the capacity profile of DOD100%. Consequently, the electrodes undergo the abrupt a–c phase transformation under DOD100% and mere a–a volume changes under DOD70–90% even at 1 C (Figs. 2e, f). Extra capacity decay due to the repeated c-Li$_{3.75(+\delta)}$Si formation/decomposition under DOD100%[7,26,30,41] is discussed in Methods under "Reference electrochemistry" and Supplementary Fig. 1. Relative measurement errors of CE are below ±0.1% as shown in Supplementary Fig. 2, by using reasonable cycler

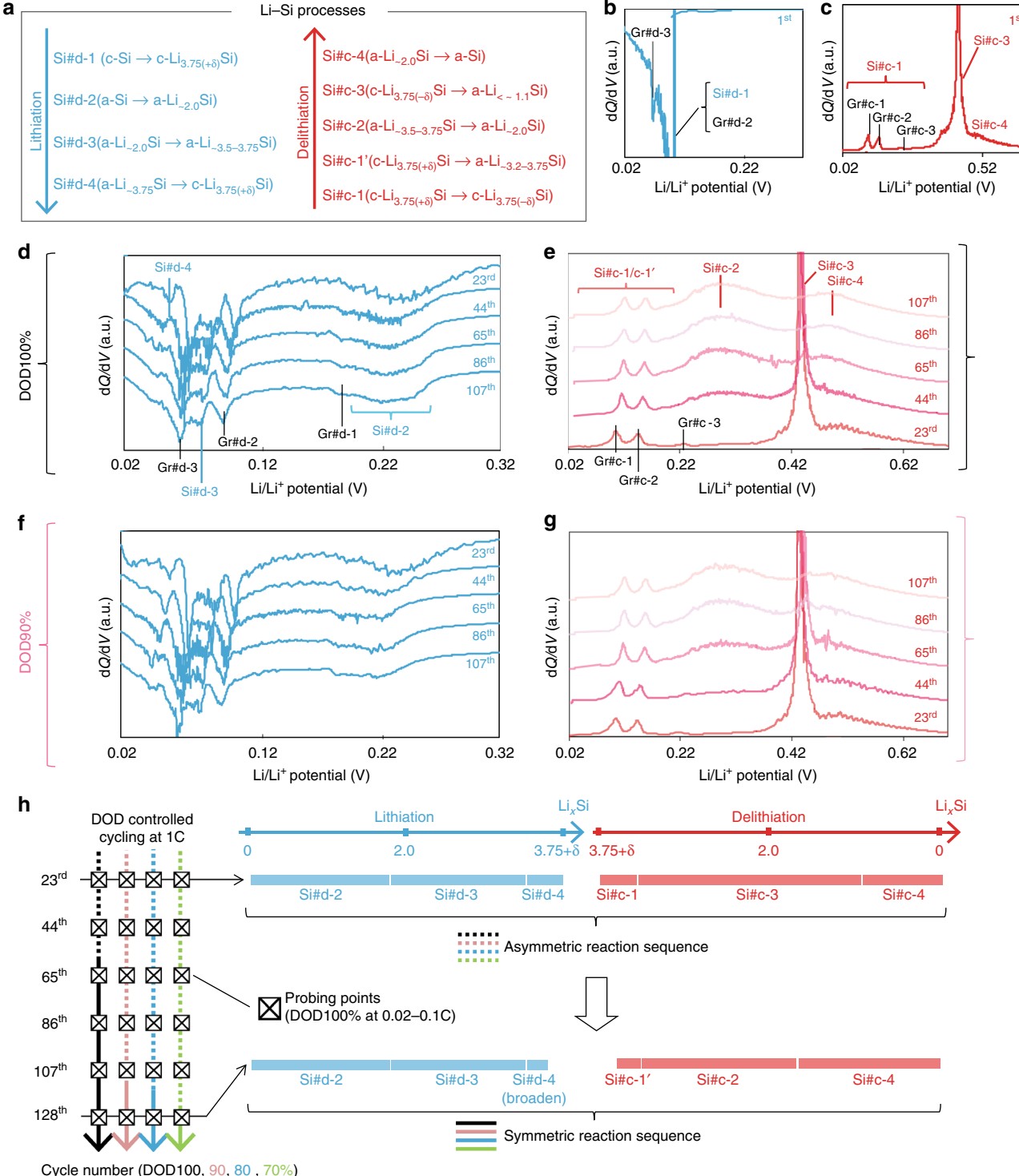

**Fig. 3** Evolution of Li–Si electrochemical processes over cycling under different DOD controls. **a** Notation of the stoichiometric Li–Si electrochemical processes on (de)lithiation. Si#d-X and Si#c-X are Li–Si processes, while #d-X and #c-X denote the $X^{th}$ discharge (lithiation) and charge (delithiation) processes in the half-cells, respectively. For corresponding Gr#d-X and Gr#c-X processes, see Supplementary Fig. 3. The notations are also summarized in Methods (under "Reference electrochemistry") and Supplementary Table 5. $dQ/dV$ profiles for type-A electrodes in the first (**b**) lithiation (discharge) and (**c**) delithiation (charge). $dQ/dV$ profiles at the probing points (every 20 cycles at 0.1 C under DOD100%) on (**d, f**) lithiation and (**e, g**) delithiation for (**d, e**) DOD100% and (**f, g**) DOD90%, respectively. The $dQ/dV$ profiles are stacked with a constant pitch to show the different processes more clearly. **h** Schematics of change in the electrochemical Li–Si process flow at the probing points during (de)lithiation over cycling under different DOD controls. The stoichiometry of each process is indicated by the length of each bar on lithiation (blue bars) and delithiation (red bars) as a function of Li concentration in $Li_xSi$. The reaction flow in the earlier cycling stage is asymmetric (dotted lines), while that in the later stage is symmetric (solid lines). The transition from the asymmetric to symmetric regimes is clearly accelerated by undergoing DOD100% protocol. The corresponding data for type-B electrode are shown in Supplementary Fig. 4 with the same tendency as in here

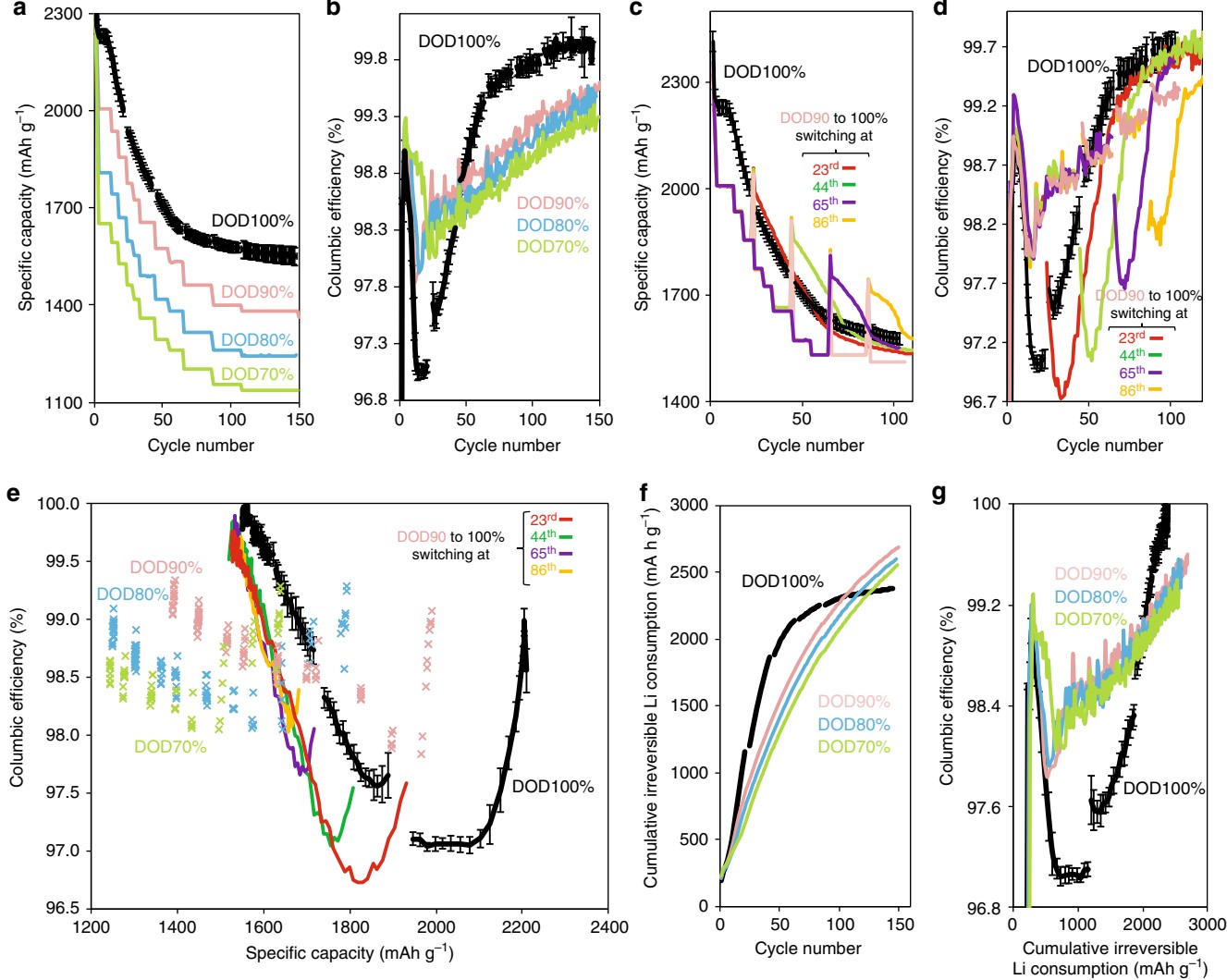

**Fig. 4** Baseline electrochemical properties under different lithiation depth controls. Electrochemical properties for type-A electrodes under different depth of discharge (DOD) controls. **a** Specific capacity and (**b**) Coulombic efficiency (CE) for different DOD controls over 150 cycles. **c** Specific capacity and (**d**) CE for DOD90% switched to DOD100% at different cycle numbers; the switching points for red, green, purple, and yellow solid lines are the 23rd, 44th, 65th, and 86th cycles, respectively. **e** CE as a function of the specific capacity for different DOD controls. Colored solid lines (red, green, purple, and yellow) show the profiles switched from DOD90% to DOD100% in (**c**, **d**). CE is clearly the highest under DOD100% after a certain loss of reversible capacity. **f** Cycle number plotted over accumulated irreversible Li consumption. The cumulative consumption is the lowest among the given DODs after certain cycles. **g** CE plotted over accumulated irreversible Li consumption. CE under DOD100% protocol is the highest among the given DOD controls after a certain amount of sacrificial Li consumption. Note that the CE error bars for DOD70–90% are all below ±0.1% and therefore omitted in (**b**, **d**, **e**, **g**). The corresponding results for type-B electrode are shown in Supplementary Fig. 13

calibration and reproducible electrode fabrication techniques (Methods under "Accuracy of cycler").

**Li–Si electrochemical processes for different DOD pathways.** To clarify the Li–Si processes on (de)lithiation, we use the following notation, which is also used in the previous study[25]. Si#d-X and Si#c-X are Li–Si processes, while #d-X and #c-X denote the Xth discharge (lithiation) and charge (delithiation) processes in the half-cells, respectively (Methods under "Reference electrochemistry"). The Li–Si processes (Fig. 3a) are interpreted from the dQ/dV profiles (Gr#d-X and Gr#c-X are Li–graphite (Gr) ones, separately examined in Supplementary Fig. 3). The author's previous work[25] showed correlations between Si#d-X and Si#c-X[25]. In the 1st cycle, Si#d-1 (100 mV, c-Si → c-Li$_{3.75(+\delta)}$Si) and Si#c-3 (430 mV, c-Li$_{3.75(-\delta)}$Si → a-Li$_{<\sim1.1}$Si) are correlated. In the

following cycles, the correlation becomes more symmetric when cycled between amorphous Li$_x$Si components; Si#d-2 (250–300 mV, a-Si → a-Li$_{\sim2.0}$Si) coupled with Si#c-4 (550 mV, a-Li$_{\sim2.0}$Si→a-Si), and Si#d-3 (100 mV, a-Li$_{\sim2.0}$Si→ a-Li$_{3.5-3.75}$Si) with Si#c-2 (300 mV, a-Li$_{3.5-3.75}$Si→a-Li$_{\sim2.0}$Si). In contrast, it does more asymmetric when the a–c phase transformation occurs: Si#d-4 (50 mV, a-Li$_{\sim3.75}$Si → c-Li$_{3.7(+\delta)}$Si) and Si#c-3.

Figures 3b–g show evolution of the dQ/dV profiles for type-A electrodes at the amorphization process (the 1st cycle, Figs. 3b, c) and at the probing points under DOD100 (Figs. 3d, e) and 90% (Figs. 3f, g). Indeed in this study, the first cycle is dominated by Si#d-1/Si#c-3 correlation . For the subsequent 20 cycles under DOD100% protocol, the processes are dominated by the asymmetric Si#d-4/Si#c-3 correlation. The correlation gradually becomes symmetric after the 40th cycle; Si#c-3 significantly decreases with the broadening of Si#d-4 and increase of Si#c2.

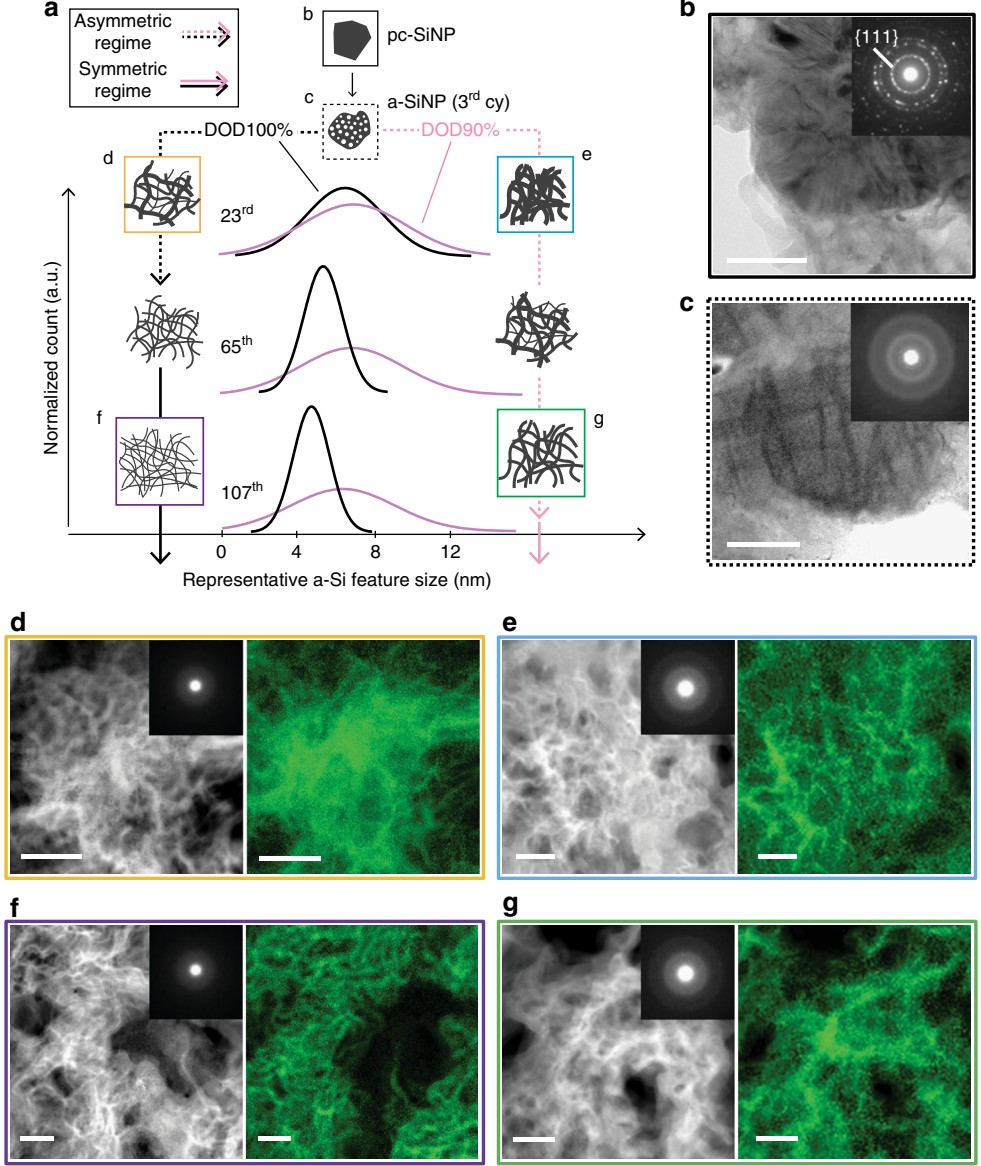

**Fig. 5** Size distribution and electron microscopy images of fully delithiated Si. **a** Schematics of Si morphological changes over 107 cycles, combined with the Gaussian-fitted size distribution curves of delithiated amorphous-Si (a-Si) for type-A electrodes cycled under depth of discharge (DOD)100% (left) and 90% (right). The distribution curves are stacked with a constant pitch to show the changes over cycles more clearly. Dotted and solid lines show the electrochemical asymmetric and symmetric regimes (Fig. 3h), respectively. The schematics surrounded by different colored squares correspond to the following electron microscopy images. Bright-field TEM images for (**b**) as-is polycrystalline silicon nanoparticles (pc-SiNP) and (**c**) porous amorphous SiNP (a-SiNP) after the 2nd cycle. Inset shows the selected area diffraction pattern (SADP) of the imaged area. **d–g** High-angle annular dark field (HAADF) with SADP inset (left) and electron energy loss spectroscopy (EELS) mapping (right) of delithiated a-Si at (**d, e**) the 23rd and (**f, g**) the 107th cycle for (**d, f**) DOD100% and (**e, g**) DOD90%, respectively. All scale bars are 50 nm. Combining the results from HAADF and EELS, the feature size distribution for DOD100% becomes much narrower compared to that for DOD90% after the 65th cycle. The trend of particle size and distribution is also shown in Supplementary Fig. 15

Eventually, Si#c-3 is overshadowed by Si#c-2 and Si#c-4 after the 65th cycle, and the processes are consequently dominated by the symmetric Si#d-2/Si#c-4 and Si#d-3/Si#c-2 correlations (Fig. 3h). Notably, Si#c-3 disappears despite the presence of a broadened Si#d-4, i.e., a contradiction to the Si#d-4/Si#c-3 correlation. Hence, we tentatively assign a new electrochemical process, Si#c-1' (c-Li$_{3.75(+\delta)}$Si → a-Li$_{\sim3.2-3.75}$Si) at 10–150 mV (overlapping Gr#c-1 and Gr#c-2), which is further examined by the following XRD and NMR analyses. The same trend is observed in type-B electrodes (Supplementary Fig. 4). Effects of current rates (Supplementary Fig. 5–7) and Li-metal resistance (Supplementary Fig. 8–10) on the Li–Si processes are considered in Methods

under "Reference electrochemistry" and "Li-metal resistance", respectively. These results indicate that the asymmetric-to-symmetric shift in the d$Q$/d$V$ profiles are not simply due to kinetics, but to altered energetics in the Li–Si reaction paths. In contrast, when DOD is controlled to 70–90%, Si#c-3 completely disappears only after ~107–140 cycles (Figs. 3f, g). Thus, repeating the phase transformations under DOD100% drastically accelerates the asymmetric-to-symmetric shift (Fig. 3h).

**Coulombic reversibility of Li–Si processes.** CE alterations under different DOD controls are investigated over ~190 cycles for type-A electrodes (Fig. 4). Over the first 20 cycles, CE under

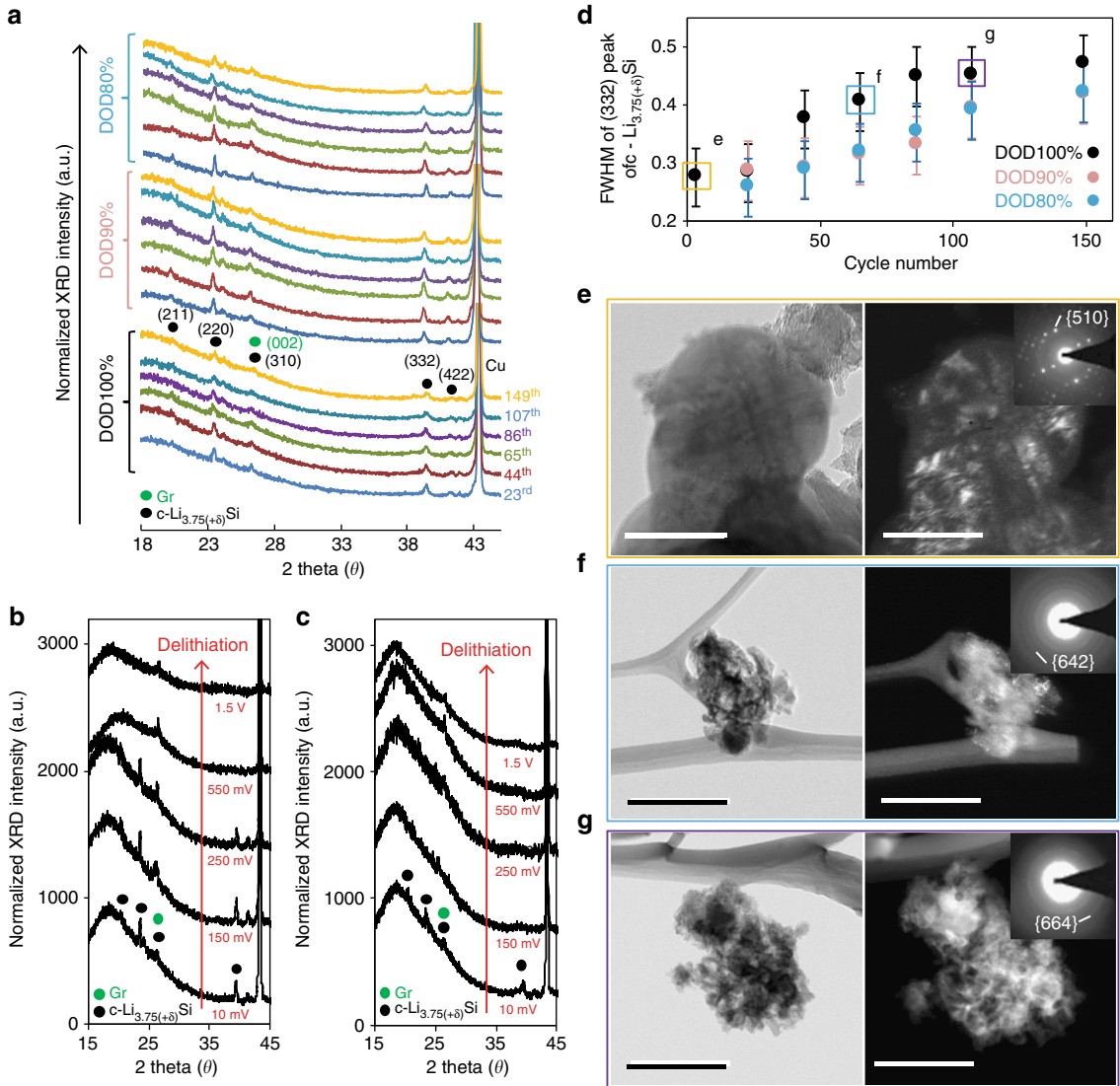

**Fig. 6** XRD profiles and electron microscopy images of fully lithiated Si. **a** XRD spectra of fully lithiated Li–Si alloys for type-A electrodes at 10 mV for depth of discharge (DOD)100, 90, and 80% over 150 cycles. XRD spectra at various delithiation potentials under DOD100% at (**b**) the 23$^{rd}$ cycle and (**c**) the 86$^{th}$ cycle. In the 23$^{rd}$ cycle, c-Li$_{3.75(+\delta)}$Si diffraction peaks disappear only after 550 mV on delithiation, while it does so at 150 mV at the 86$^{th}$ cycle. **d** Full width of half maximum (FWHM) of the (332) peak in c-Li$_{3.75(+\delta)}$Si XRD profiles over 150 cycles for DOD100, 90, and 80%. FWHM for DOD100% saturates near 65$^{th}$ cycle, while that for DOD90% and 80% continuously increases until the 150$^{th}$ cycle. Empty squares colored yellow, blue, and purple correspond to fully lithiated electrodes at 10 mV shown in the following electron microscopy images. **e**–**g** Bright-field and dark-field TEM images with selected area diffraction pattern (SADP) and index planes shown in inset. The fully lithiated Si in (**e**) the 3$^{rd}$ cycle, (**f**) the 65$^{th}$ cycle, and (**g**) the 107$^{th}$ cycle under DOD100% controls. Scale bars are (**e**) 200 nm and (**f**, **g**) 500 nm. It is obvious that c-Li$_{3.75(+\delta)}$Si crystals become much smaller in the 65$^{th}$ and the 107$^{th}$ cycles, compared to those in the 3$^{rd}$ cycle, which agrees with the trend seen in (**d**)

DOD100% shows an abrupt decrease down to 96.5% (Figs. 4a, b). This is followed by a sudden increase in CE during the 20$^{th}$–60$^{th}$ cycles, reaching a saturation value of around ~99. 9% after about 80 cycles with ~23% capacity loss. In contrast, under DOD70–90%, the initial CE decrease is less severe, while its subsequent increase is more moderate, reaching ~99.7–99.9% only after 150–200 cycles. Interestingly, the times, at which CE reaches ~99.5% and CE profile becomes more stable, are synchronized with the Li–Si electrochemical regime shift from asymmetric to symmetric. The CE profiles for a–a volume changes from DOD70–90% exhibit a constant difference. However, the profile abruptly changes from DOD90% to 100%. These results indicate that the reversibility is correlated with the repeated c-Li$_{3.75(+\delta)}$Si formation/decomposition and the associated electrochemical regimes.

For a more detailed examination, the electrodes under DOD90% are cycled for $Y$ cycles ($Y$ = 22, 43, 64, 85, and 106) and then abruptly switched to DOD100% from the $(Y+1)^{th}$ cycle onward (Fig. 4c), in order to repeat the c-Li$_{3.75(+\delta)}$Si formation/decomposition. Upon switching, the CE undergoes a sudden decrease followed by a rapid increase for all $Y$ values, as shown in Fig. 4d. The depth of these CE drops becomes shallower, and their width narrower as $Y$ increases. Also, the number of cycles required to reach saturation CE value decreases as $Y$ increases. As shown in Figs. 3f, g, the electrode under DOD90% gradually shifts from asymmetric to symmetric after 107$^{th}$ cycles. These results indicate that the duration of the remaining asymmetric regime in a given electrode system determines the behavior of CE in the following cycles under the given DOD controls. In other words, CE becomes

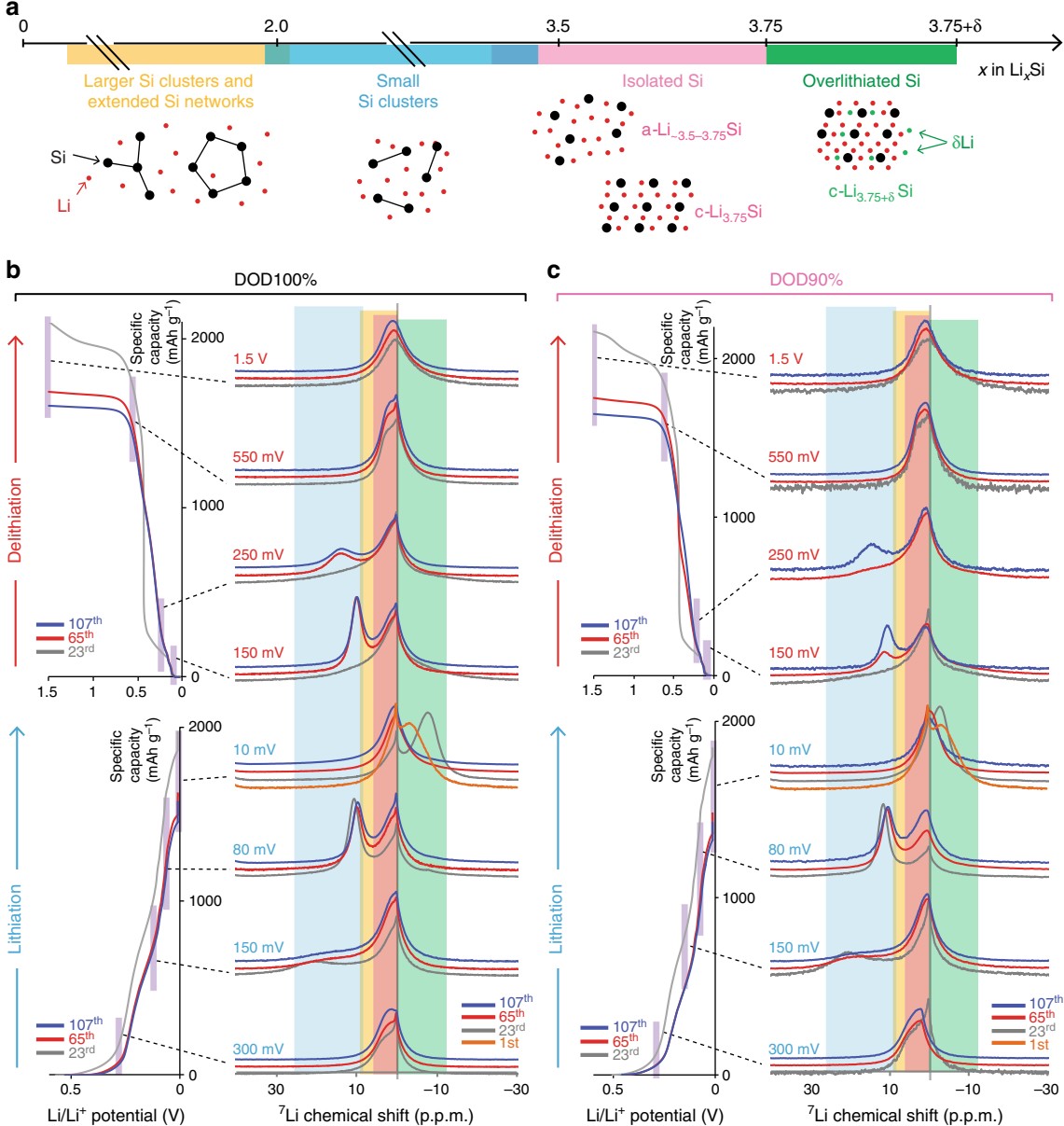

**Fig. 7** Ex situ MAS $^7$Li solid-state NMR spectra at various (de)lithiation potentials. **a** Schematics of various Li–Si local environments as a function of Li concentration in Li–Si alloys. The $^7$Li resonances are highlighted by yellow (10–0 p.p.m., larger Si clusters and extended Si networks), blue (25–10 p.p.m., small Si clusters), pink (6–0 p.p.m., isolated Si$^{4-}$ anions including c-Li$_{3.75}$Si), and green (0 to −10 p.p.m., overlithiated crystalline phase, c-Li$_{3.75+\delta}$Si). The notations for the Li–Si environments and their correlation with the Li–Si electrochemical processes are also summarized in Supplementary Table 5. The recorded spectra at various Li/Li$^+$ potentials over 107 cycles at the probing points under (**b**) depth of discharge (DOD)100% and (**c**) DOD90%. The NMR spectra at each potential are linked to the corresponding points in the capacity–voltage profiles. Orange, gray, red, and blue solid lines correspond to the spectra at the 1$^{st}$, 23$^{rd}$, 65$^{th}$, and 107$^{th}$ cycles, respectively. The spectra for different cycle numbers and different potentials are stacked with constant pitch to show the different processes more clearly

higher and less susceptible to the presence of c-Li$_{3.75(+\delta)}$Si as Li–Si processes get closer to the symmetric one.

So as to further highlight the irreversibility nature, CE is plotted under the same reversible capacity for the various DOD cycling protocols in Fig. 4e. These plots show that, when compared under the same capacities, CE can be significantly altered by the Li–Si electrochemical pathways during previous cycles. Interestingly, the pathway under DOD100% exhibits the highest reversibility trajectory for all given DOD protocols when the residual capacity gets below ~77–80%. Importantly, this shows that the prominent CE increase seen in DOD100% is not simply due to capacity decay, but rather to altered nature of the

electrodes granted by the repeated a–c phase transformations (details in Methods under "Reference electrochemistry").

Further, cumulative irreversible Li consumption over cycles is explored for different DOD controls (Figs. 4f, g). The consumption under the DOD100% protocol is the lowest among all cases after 70–100 cycles (Fig. 4f). Also, the CE profile plotted over the cumulative Li consumption under DOD100% clearly underlies those for the other DOD controls (Fig. 4g) after a certain amount of Li consumption. Importantly, these profiles under DOD100% take clearly different trends from those under DOD70–90%. These findings indicate that the repeated a–c phase transformations could minimize the cumulative irreversible Li consumption

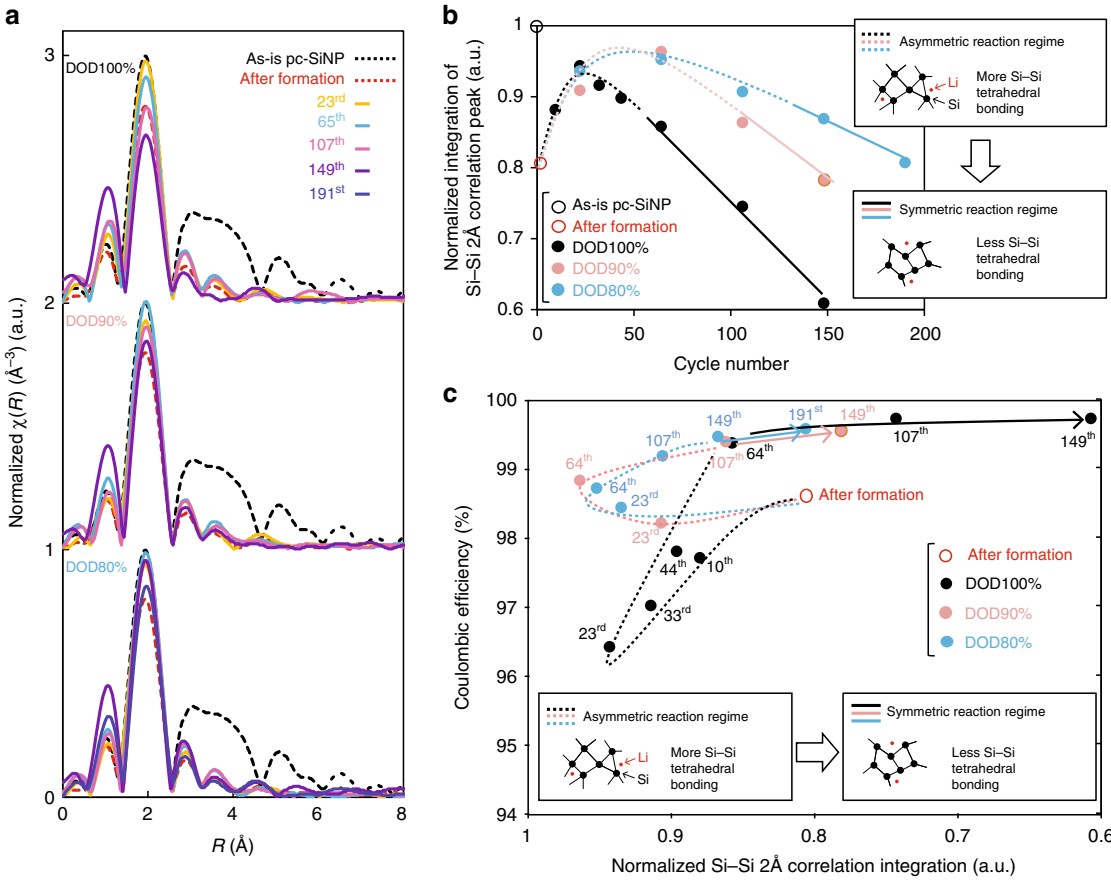

**Fig. 8** Ex situ XAFS analyses for delithiated amorphous-Si. **a** Stacked Fourier-transformed EXAFS profiles at Si K-edge for fully delithiated type-A electrodes at 1.5 V over 190 cycles under depth of discharge (DOD)100, 90, and 80%. Black and red dotted lines, and yellow, blue, magenta, purple, and blue solid lines correspond to the as-is electrodes, the electrodes after the amorphization of c-Si, and after the 23rd, 65th, 107th, 149th, and 191st cycles, respectively. The profiles for the same DOD control are overlaid, whereas those under different DOD controls are stacked with a constant pitch to show the intensity changes more clearly. **b** Normalized integration of 2 Å Si–Si correlation peak in (**a**), named $A_{(2Å Si-Si)}$, over 190 cycles for different DOD controls. The as-is type-A electrodes and the electrodes after amorphizing pc-SiNP are shown by black and red empty circles, respectively. The dashed and solid lines indicate the asymmetric and symmetric regimes seen in the Li–Si electrochemical processes (Fig. 3h), respectively. Schematized Si–Si tetrahedral boding environments are illustrated in the chart, suggesting the regime shift from asymmetric to symmetric. **c** Coulombic efficiency (CE) as a function of $A_{(2Å Si-Si)}$ over 190 cycles for different DOD controls

for mid to longer-term cycles. Notably, although c-Li$_{3.75(+\delta)}$Si is commonly recognized as a degradation factor, we uncovered a positive aspect of it for the first time. One issue to be carefully considered is differences in the exposure time of Si to the electrolyte at constant voltage (CV) domains on lithiation for different DOD controls (see Methods under "Reference electrochemistry" and Supplementary Fig. 11, 12). Since the electrochemical processes and the reversibility trends for type-B electrode (Supplementary Fig. 13) are the same as those for type-A, only type-A electrodes are discussed in all the following structural analyses.

**Morphological analysis via electron microscopies**. Ex situ TEM imaging highlights the morphological change of delithiated Si under different DOD controls over cycles (Fig. 5a and Methods under "TEM"). The images for as-is pc-SiNP and a-SiNP after the 2nd delithiation are shown in Figs. 5b, c, respectively. The detailed imaging results of a-SiNP are also presented in Supplementary Video 1, 2 and Supplementary Fig. 14. The Gaussian-fitted size distributions of delithiated a-Si over 107 cycles for DOD90% and DOD100% are shown in Fig. 5a, with the corresponding schematic a-Si morphologies. Dark-field TEM images combined with elemental mapping via electron energy loss spectroscopy are

shown in Figs. 5d–g. Over the first 23 cycles, the amorphized spherical structures drastically change: they expand and merge with each other, resulting in widespread three-dimensional networked structures for both DOD100 and 90% (Figs. 5d, e). For 24–107th cycle, the complex, entangled structures gradually disengage from each other and the size distribution becomes narrower to various degrees depending on the DOD controls: for DOD100%, 90%, and 80%, the average feature size ($d$) after the 107th cycle is ~4.8, 6.3, and 6.4 nm with SD ($\sigma$) of ~1.0, 2.7, and 3.0 nm, respectively (Supplementary Fig. 15). Thus, the repeated c-Li$_{3.75(+\delta)}$Si formation/decomposition under DOD100% after 107 cycles accelerates to produce smaller a-Si and narrower size distribution compared to those below DOD90%. Additional detailed images are shown in Supplementary Fig. 16–19. Ex situ XPS analysis result under DOD80–100% is presented in Supplementary Fig. 20 (details in Methods under "XPS").

**Probing c-Li$_{3.75(+\delta)}$Si crystallinity and energetics**. Since the repeated c-Li$_{3.75(+\delta)}$Si formation/decomposition governs Li–Si electrochemical processes and the irreversibility, its crystallinity is analyzed by XRD and TEM. Ex situ XRD profiles at 10 mV (details in Methods under "XRD") show c-Li$_{3.75(+\delta)}$Si reflection over ~190 cycles for DOD80–100% (Fig. 6a). The presence of c-

$Li_{3.75(+\delta)}Si$ and the following symmetric delithiation processes at 300 (Si#c-2) and 550 mV(Si#c-4) contradict the hysteretic reaction process at 430 mV (Si#c-3) in the previous lines of work[25–27,30]. To explore this inconsistency, ex situ XRD experiments are conducted at different potentials on delithiation in the asymmetric (the 23rd cycle, Fig. 6b) and symmetric (the 86th cycle, Fig. 6c) regimes. In the former, the reflection is still present at 250 mV, and it only disappears at 550 mV[25–28]. In contrast, in the

latter, it nearly disappears at 150 mV ($x > 3.2$ in a-$Li_xSi$). These results indicate that the energetics in the symmetric regime enables Li ions to de-couple from c-$Li_{3.75(+\delta)}Si$ to form a-$Li_{~3.2–3.75}Si$ below 430 mV. Thus, the tentatively defined new process Si#c-1' in Fig. 3 is rationalized. These results can rule out the possibility that the c-$Li_{3.75(+\delta)}Si$ presence in XRD reflection and subsequent absence of Si#c-3 in the dQ/dV profiles are due to electrochemically isolated c-$Li_{3.75(+\delta)}Si$.

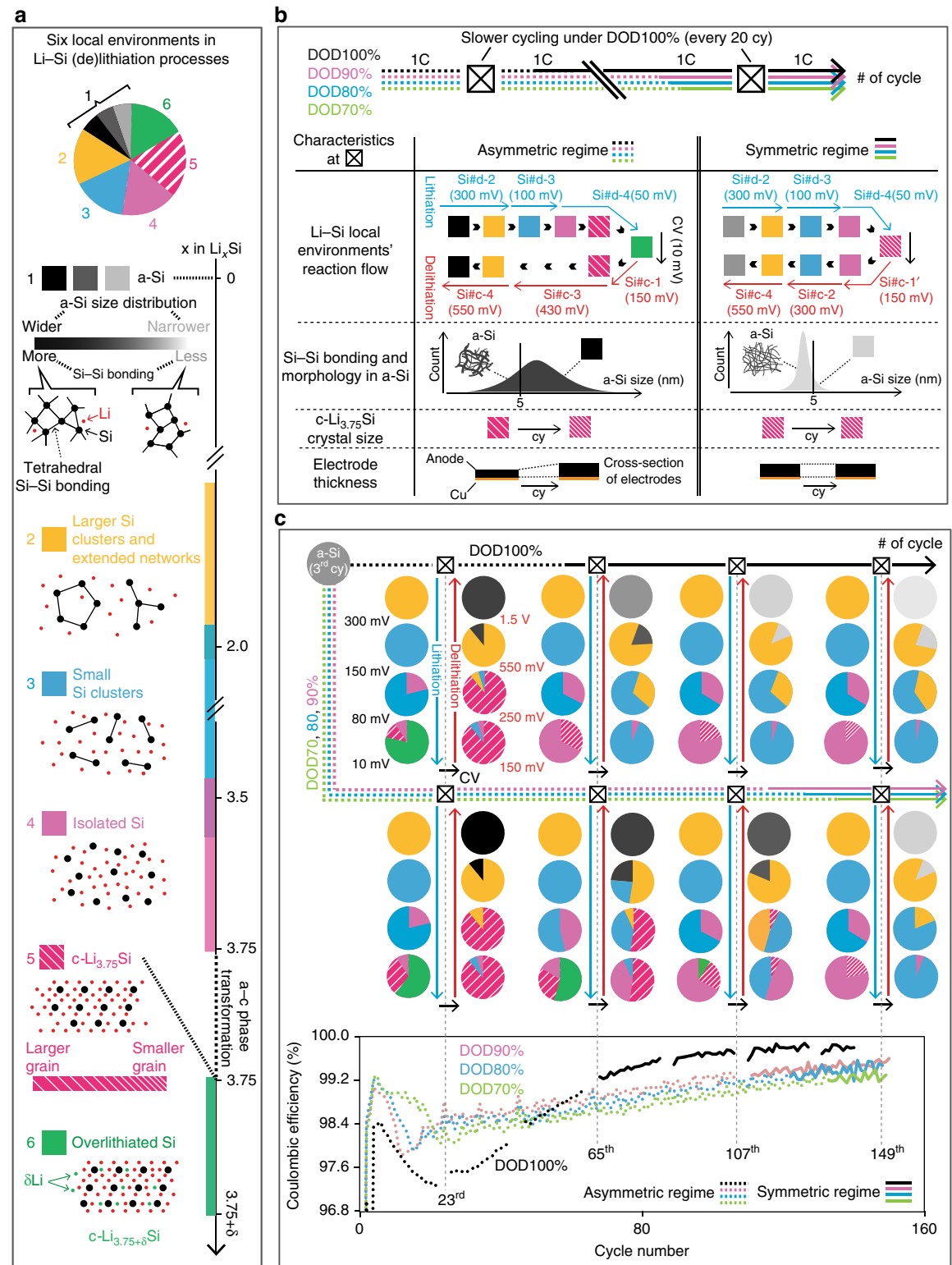

Figure 6d shows the FWHM of c-Li$_{3.75(+\delta)}$Si (332) reflection over cycles under different DOD controls. For the first 23 cycles, this value does not change significantly for all DODs, indicating that fully lithiated Li–Si alloys is large enough to accommodate a c-Li$_{3.75(+\delta)}$Si grain. From the 23$^{rd}$ cycle onward under DOD100%, it shows a steeper increase than under DOD80–90%, and the value is nearly saturated after the 86$^{th}$ cycle. In contrast, the FWHM for DOD80–90% increases slower, reaching the same level as for DOD100% only after the 150$^{th}$ cycle. Interestingly, the electrode thickness over cycle number (Supplementary Fig. 21) shows the same trend as that of FWHM. Selective area diffraction patterns for fully lithiated Si over 107 cycles under DOD100% clearly identifies bulky c-Li$_{3.75(+\delta)}$Si (dozens of nm in size) in the 3$^{rd}$ cycle and more segmented one (<5 nm) in the 65$^{th}$ and 107$^{th}$ cycles, respectively (Figs. 6e–g).

**Local structure probing via ex situ $^7$Li solid-state NMR spectroscopy.** In order to associate the Li–Si local environments with the altered electrochemical processes and CE behaviors, ex situ MAS $^7$Li ss-NMR analysis is conducted at different (de)lithiation potentials over cycles under DOD90 and 100% controls (details in Methods under "$^7$Li solid-state NMR spectroscopy"). On the basis of the previous study, the $^7$Li resonance is assigned as shown in Fig. 3a[25]: resonance at ~20–10 p.p.m. corresponds to $^7$Li near small Si clusters; that at 10–0 p.p.m. to larger Si clusters and extended Si networks, isolated Si anions including c-Li$_{3.75}$Si; and that at −10 p.p.m. to overlithiated crystalline phase c-Li$_{3.75+\delta}$Si. These environments are associated with the electrochemical processes in Supplementary Table 5.

In the 23$^{rd}$ cycle, the local environments during (de)lithiation evolve under the asymmetric sequence for both for DOD100% and 90% (Figs. 3b, c). On lithiation, a-Si atoms are gradually broken into smaller Si clusters, ending up with isolated Si atoms and overlithiated Si component. The environments do not symmetrically reform Si–Si small clusters on delithiation, but asymmetrically form larger Si clusters and networks. Key et al. and Ogata et al. showed that residual Si clusters or defects in c-Li$_{3.75(+\delta)}$Si structure dominantly serve as nuclei and control the hysteretic energetics on delithiation to grow amorphous-Si matrix[23–25]. This is because such reactions are more cost-friendly than migrating highly charged isolated Si anions close together and re-forming small Si clusters. In contrast, after the 65$^{th}$ cycle under DOD100%, the spectrum at 10 mV is dominated by isolated Si anions without +δ component at −10 p.p.m. Combined with the XRD results, this finding shows that isolated Si anions form c-Li$_{3.75}$Si without proceeding to +δ environments despite more favorable energetics for forming c-Li$_{3.75+\delta}$Si than breaking the residual Si–Si bond[25]. Notably, the profile at 150 mV

on delithiation well overlaps with that at 80 mV on lithiation, indicating that most of isolated Si anions in c-Li$_{3.75}$Si can reversibly reform small Si clusters. These results also rationalize the newly assigned process Si#c-1'. In contrast, the spectra for DOD90% in the 65$^{th}$ cycle still show the asymmetric reaction flow of the environments. The sequence only becomes more symmetric after the 107$^{th}$ cycle, which is also synchronized with the electrochemical regime shift.

**Probing local structure of a-Si via ex situ XAFS.** In order to probe fully delithiated a-Si local environments, ex situ XAFS (X-ray absorption near-edge structure (XANES) and extended X-ray absorption fine structure (EXAFS)) analyses are conducted at the Si K-edge (details in Methods under "XAFS"). The XANES profiles are summarized in Supplementary Fig. 22a. The EXAFS profiles (Supplementary Fig. 22b) are extracted from the XAFS data and converted into Fourier-transformed profiles, which are relevant to the radial distribution function (RDF) of Si. Figure 8a clearly shows that Si is fully amorphized to leave only Si–Si tetrahedral correlations (2 Å). Normalized integrations of 2 Å Si–Si correlations peaks, named $A_{(2\text{Å Si–Si})}$, are used to index tetrahedral Si–Si bonding environment. The data of $A_{(2\text{Å Si–Si})}$ over cycles are shown in Fig. 8b. After the 2$^{nd}$ cycle, $A_{(2\text{Å Si–Si})}$ suddenly decreases to ~0.81, which is in line with the initial marked structural change into a complex porous sphere (Supplementary Fig. 14 and Supplementary Video 1, 2). Over the next dozens of cycles, $A_{(2\text{Å Si–Si})}$ temporarily increases for all DODs to reach a local maximum with different timings for each DOD. After reaching the local maximum, $A_{(2\text{Å Si–Si})}$ linearly decreases with different gradients for different DOD protocols. The repeated phase transformations under DOD100% accelerate changes in environments in the primary Si–Si bonding. The temporal $A_{(2\text{Å Si–Si})}$ increase and the following decrease correspond to the agglomeration and the following disengage of the a-Si structures over cycles seen in the TEM imaging.

Interestingly, when CE is plotted over $A_{(2\text{Å Si–Si})}$, the curves in Fig. 8c are characteristically constrained by the different DOD protocols. When $A_{(2\text{Å Si–Si})}$ increases from ~0.81 to 0.95 as the cycle proceeds, CE decrease is significantly susceptible to presence of c-Li$_{3.75(+\delta)}$Si on (de)lithiation. In this period, the profile for DOD100% can deviate from that for DOD80–90% by ~1.5%. Once $A_{(2\text{Å Si–Si})}$ reaches up to ~0.95 over cycles, it starts to decrease with CE increase over cycles for all DODs. The CE increase rate is significantly influenced by a given DOD; CE for DOD100% quickly increases from 96.4 to 99.5% as $A_{(2\text{Å Si–Si})}$ decreases from 0.94 to 0.86 before the 65$^{th}$ cycle, while that for DOD80–90% shows a gentler increase, reaching ~99.5% only after 149 cycles. Interestingly, regardless of DOD controls, when

**Fig. 9** Schematics of the correlation between Li–Si electrochemical/structural characteristics and Coulombic efficiency. **a** Legend of six different Li–Si local environments upon (de)lithiation, which are used in (**b**, **c**). Fully delithiated amorphous (a-)Si and c-Li$_{3.75}$Si are further subcategorized, according to the electrochemical and various atomistic structural probing results listed in Fig. 1. **b** Schematics of structural characteristics seen in two different electrochemical Li–Si reaction regimes. The dotted and solid arrowed axes at top show the asymmetric and symmetric regimes over cycle, respectively, the duration of which is significantly altered by the depth of discharge (DOD) control. The crossed empty squares embedded in the lines show the structural probing points inserted every 20 cycles. The shifts of the structural characteristics seen at these points in each regime are synchronized with the electrochemical regime shift. Firstly, the reaction flow of the Li–Si local environments also changes from asymmetric to symmetric, which is overlaid with the electrochemical processes. Secondly, the properties of Si–Si bonding and the morphology in a-Si are altered. Thirdly, the c-Li$_{3.75}$Si crystal size is saturated at the shift. Finally, the electrode thickness also saturates near the shift. **c** Profiles of the local Li–Si environments at various potential during (de)lithiation over cycling under different DOD controls. The phases formed on lithiation and delithiation are shown on the left and right sides along blue and red arrows, respectively. The constant voltage domain at 10 mV is illustrated by black arrows. Each component in the pie charts shows a local environment present at the given potential, while their sizes indicate the relative proportion of the phases. The dotted and solid lines in cycle number and CE profiles indicate the asymmetric and symmetric regimes, respectively. The local environments and their profiles on (de)lithiation significantly depend on the affiliated electrochemical regimes, which consequently governs CE profiles. The affiliation and CE behaviors can be prominently altered by the repeated a–c phase transformations

$A_{(2Å \, Si–Si) >} \sim 0.8–0.85$, CE is susceptible to the DOD controls, while it is not when $A_{(2Å \, Si–Si) <} \sim 0.8–0.85$. Notably, as indicated by the dotted/solid lines in Fig. 8c, the time at which the CE susceptibility changes near $A_{(2Å \, Si–Si)} \sim 0.8–0.85$ is synchronized with the electrochemical asymmetric-to-symmetric shift under all DOD controls.

## Discussion

In this section, we discuss the mechanism behind the correlations between the altered structural/electrochemical characteristics under different DOD controls and the Coulombic reversibility behaviors. This is schematically summarized in Figs. 9a–c. Figure 9a explains the legend schematics of six different Li–Si local environments upon cycles, ordered according to the Li concentration. a-Si and c-Li$_{3.75}$Si are further subcategorized by morphology/bonding states and c-Li$_{3.75}$Si grain size, respectively.

Figure 9b highlights four key structural characteristics that change when the electrochemical regime shifts from asymmetric to symmetric. Remarkably, the changes are explicitly synchronized with the asymmetric-to-symmetric electrochemical regime shift. Firstly, a reaction flow of the local environments on (de)lithiation also changes from asymmetric to symmetric. Along the shift, c-Li$_{3.75}$Si gradually stops forming the +δ component (c-Li$_{3.75+δ}$Si) on lithiation, and it starts to symmetrically reform small Si clusters on delithiation. The transition on (de)lithiation undergoes with the six different Li–Si environments. This is very different from the flow of the hysteretic environments seen in the asymmetric regime; the transition does with only the five environments. Secondly, a-Si structure is more dominated by surface with smaller and more uniform Si features. Interestingly, the regime shift always occurs when the normalized index of a-Si 2 Å correlations, $A_{(2Å \, Si–Si)}$, falls below ~0.8–0.85, regardless of previously used DOD controls (Figs. 8b, c). Further, the change in c-Li$_{3.75(+δ)}$Si crystal size in the asymmetric regime becomes relatively constant, which is in line with a trend seen in electrode thickness over cycles (Supplementary Fig. 21).

Our preliminary DFT calculation (Supplementary Fig. 23) provides some explanations for the absence of +δ. The formation and surface energies in bulk Si and 2-nm Si clusters are calculated along Li concentration in the alloy (details in Methods under "Numerical calculations"). The driving force for lithiating a-Li$_x$Si beyond $x = 3.25$ is significantly lower in the nanocluster compared to that in bulk. This is probably owing to an increased contribution of the high surface energy to the total formation energy (FE) in the surface-dominated system. In such a system, Li atoms might be less prone to inhomogeneously overlithiate c-Li$_{3.75}$Si nuclei near the surface. Instead, breaking the residual Si–Si bonds is more preferred, resulting in more uniform lithiation and much less +δ at the end of lithiation. The capability for c-Li$_{3.75}$Si to symmetrically reform small Si clusters on delithiation may be attributed to the altered energetics in a more surface-dominated system, in which the energy barrier to remove Li atoms out of the c-Li$_{3.75}$Si matrix could become much lower than that in bulk. The saturation trends in c-Li$_{3.75(+δ)}$Si crystal size and electrode thickness are probably due to less destructive or more efficient stress release processes in the symmetric regime. The origin of the accelerated changes in the electrochemical/structural characteristics via repeating c-Li$_{3.75(+δ)}$Si formation/decomposition probably lies in the ability of c-Li$_{3.75(+δ)}$Si to rearrange the Si–Si primary bonding on delithiation and in consequent formation of the uniform surface-dominated system with potentially altered interface property.

These changes of structural characteristics upon the regime shift are further resolved at various potentials at the probing points in Fig. 9c. Profiles of the local components at various

potentials are illustrated in the form of pie, the flow of which is further linked to the CE profiles at the bottom. The asymmetric/symmetric regime is indicated by dotted/solid lines, respectively, in the cycle number and CE profiles. The changes in the profile transition along the regime shift are clearly associated the susceptibility nature of CE. CE significantly fluctuates when the profile development on (de)lithiation undergoes the complex asymmetric interplays among the six Li–Si environments while CE becomes much more stable when it does more symmetric ones among the five environments. The accelerated shift of electrochemical/structural characteristics and of the CE behaviors by repeated c-Li$_{3.75(+δ)}$Si formation/decomposition under DOD100% is clearly distinguished from that during incremental a–a volume changes under DOD70–90%. Combining all the results, the altered CE behavior can be explained by the following. First, the absence of +δ component in c-Li$_{3.75}$Si matrix (which is potentially a strong reduction source)[23,24] and more uniform Li concentration in the system may result in more uniform passivation and consequently reduce the irreversible formation of byproducts. Secondly, a more reversible c-Li$_{3.75}$Si formation/decomposition, enabled by the altered Li–Si energetics and the surface-dominated system, may reduce the number of irreversibly trapped/isolated Li atoms in the structure. Thus, along the regime shift, the Li–Si environments that are susceptible to the irreversible consumption either disappear or change their energetics and become more inert.

Our study provided evidence that the repeated c-Li$_{3.75(+δ)}$Si formation/decomposition over cycles, which is typically featured as a major degradation factor in the anode, indeed has an ability to inherently empower the anodes, improving CE and minimizing cumulative irreversible Li consumption. The insights can open up new possibilities for Si-rich anodes with new designs. For example, re-defining the anode/cathode capacity ratio with optimal pre-lithiation dose in the anodes, it might be possible for the anodes to undergo repetitive c-Li$_{3.75(+δ)}$Si formation/decomposition even in a full cell and consequently in situ deplete the irreversibility in the anode. Importantly, the insight can be applied to not only next-generation Li-ion batteries, but also Li–sulfur and Li-metal batteries with solid/liquid electrolytes.

## Methods

**Baseline active materials.** Active materials in the form of secondary particles are synthesized by conventional spray-drying method (B290 Mini Spray-dryer, Buchi). The secondary particles are designed to have porous open structures that can be easily wetted by the electrolyte and undergo relatively abrupt phase transformations even at higher current rates (e.g., 1 C). These secondary particles are composed of defective pc-SiNP (Stream-Si, ~120 nm), MWCNT (15 nm, CNT Co. Ltd.) with/without flake-type Gr (FT-Gr, SPG1, SEC carbon), and polyvinyl alcohol (PVA, Sigma-Aldrich, MW ~50 k). First, the components are dispersed in DI water, followed by 2 h of ultrasonication. The SiNP/MWCNT/FT-Gr/PVA ratios are 55.9/7.8/34.3/2.0 and 87.0/10.8/0/2.2 (wt%) for type-A and -B secondary particles, respectively (Supplementary Table 4). The dispersed slurry is then spray-dried with a two-fluid-type nozzle at an inlet temperature of 220 °C in 60 standard liter per min N$_2$ flow, with subsequent thermal treatment at 900 °C for 5 h in an N$_2$ atmosphere (100 °C/h ramping rate), followed by sieving (<32 μm) to remove larger secondary particles. These processes create well-defined physical parameters for the secondary particles. The average secondary particles have a diameter of ~10 μm with specific surface areas of 29.5 and 39.5 m$^2$ g$^{-1}$ for type-A and -B, respectively. SiNP and MWCNT are well entangled in the secondary particles to secure good electrical connections and buffer space to accommodate the volume expansion of Si (Figs. 2c, d). Gr in type-A secondary particles acts as a scaffold to maintain the spherical shape and electrical contacts (Fig. 2c). The wt% of Si in type-A and -B secondary particles is quantified by inductively coupled plasma spectroscopy (Shimadzu quartz torch, Nebulizer-Meinhard-type glass) to be ~54–55 and ~86–87 wt%, respectively, in good agreement with tabulated values after calcination, assuming that the wt% of residual carbon from PVA after the calcination is ~20%.

The initial reversible theoretical capacities of Si, MWCNT, and Gr are taken to be 3818 (3579 + 239, assuming δ ~0.2–0.3 in c-Li$_{3.75+δ}$Si)[25], 200, and 350 mAh g$^{-1}$[42], respectively, giving theoretical capacities of 2273 and 3342 mAh g$^{-1}$ for type-A and -B particles, respectively. These values are in good agreement with the

experimentally determined initial reversible capacities (~2250 and ~3350 mAh g$^{-1}$) to ~99% accuracy (Supplementary Table 5). The part of theoretical capacity due to Li–Si processes in type-A and -B active materials is calculated to exceed 94.7% and 99.5%, respectively, i.e., there are minor capacity contributions from Gr, MWCNT, and/or other components in both type-A and -B electrodes. The presence of c-Li$_{3.75 (+\delta)}$Si in the early cycling stage is clearly confirmed by the characteristic sharp peak in the d$Q$/d$V$ profiles and capacity–voltage profiles at 430 mV on delithiation (Figs. 2e, f). The presence of defects (stacking faults and twins) in pc-SiNP (Fig. 5b) plays an important role in forming c-Li$_{3.75(+\delta)}$Si in the earlier cycles even at higher current rates. This is because the defects can lower the activation energy to create c-Li$_{3.75(+\delta)}$Si from a-Li$_{3.75}$Si. This observation is in contrast to other types of Si-based anodes, in which Li incorporation into Si is limited by a protective shell[2,3,15] and/or a buffer medium[13,16] to minimize irreversible side reactions. In such anodes, Li–Si incorporation can also be kinetically limited, with $x$ in Li$_x$Si usually less than 3.75 at the end of lithiation. Consequently, these systems are dominated by a–a phase transformations in some cycles. A signature of this in half-cells is the slope-plateau at 300 and 550 mV on delithiation, corresponding to Si#c-2 (a-Li$_{\sim3.5-3.75}$Si → a-Li$_{\sim2.0}$Si) and Si#c-4 (a-Li$_{\sim2.0}$Si → a-Si), respectively (see Fig. 3a)[26,30,41]. In such anodes, it is hard to define the initial cycling points as DOD100% ($x = 3.75 + \delta$) because $x$ is typically much smaller than 3.75, and consequently the definition of DOD can be imprecise. Thus, it is also difficult to interpret whether the retention in such anodes originates from capacity sustenance or merely from balancing active material loss and the gradual activation of kinetically unused capacity[3,13,15].

Here, the importance of c-Li$_{3.75(+\delta)}$Si needs to be addressed for the following reasons. The negative/positive electrode capacity loading ratio (namely, N/P ratio) in commercial full cells is typically designed to be ~1.05 to satisfy safety constraints, i.e., avoiding Li dendrite formation on the anodes. In other words, the SOC in the anodes in these full cells during initial cycles is ~90–95%, which is equivalent to $x$ < 3.5 in Li$_x$Si if $x$ is proportional to SOC. However, c-Li$_{3.75(+\delta)}$Si can still exist when using at least the following two representative anode strategies. The first is to use Si/Gr composite electrodes, in which the good cyclability and lower volume changes of Gr ameliorate the poor cyclability and large volume changes of Si. This strategy, however, requires the electrode to be cycled at lower voltages (typically less than 60 mV)[42,43] to access the full Gr capacity. The decrease in potential is even more significant when cycled at higher current rates (e.g., >1 C) with higher current densities (mA cm$^{-2}$), owing to accelerated kinetics and the consequent inhomogeneous Li concentration, particularly along the direction perpendicular to the electrode surface[29,38]. The second strategy is to use an electrode containing a much higher proportion of Si and limit the capacity to, e.g., ~1500–2500 mAh g$^{-1}$, cutting off potentials at the higher values and cycling between different a-Li$_x$Si phases[7,41,44]. However, this strategy can still involve phase transformations, as the potential drifts down owing to capacity loss and inhomogeneity issues over cycles. Hence, the presence of c-Li$_{3.75(+\delta)}$Si is inevitable in both cases.

**Electrode fabrication and cycling conditions.** The controlled synthesis of the secondary particles via spray-drying leads to a well-controlled slurry for subsequent fabrication of electrodes with reproducible electrochemical behaviors. The electrodes are made of 79 wt% secondary particles (type-A or -B), 20 wt% polyacrylic acid (Li-PAA, Hwagyong Chemical) as a binder, and 1 wt% Kechen Black as a conductive additive. The components are mixed in a planetary mixer (Awatorirentaro, Thinky) for 15 min at 1000 r.p.m. The slurry is pasted onto a 10-μm-thick Cu foil, and the mass loading level (weighed by a Mettler Toledo XP26 instrument, ±1 μg accuracy) for type-A and -B is typically 1.8 and 1.2 mg cm$^{-2}$ (~3.0–3.5 mAh cm$^{-2}$), respectively. 2032-Type coin cells (Hohsen Corp.) are used in all the following experiments. The electrolyte is 1 M LiPF$_6$ in a 25/5/70 (vol%) mixture of fluoroethylene carbonate (FEC)/dimethyl carbonate (EC)/dimethyl carbonate (LP 30 Selectilyte, Merck). A 10-μm-thick separator (Asahi, Celguard, 1-μm-thick Al$_2$O$_3$ coated on both sides) is used. In this study, we define the electrode-specific capacity (mAh g$^{-1}$) by dividing the total capacity (mAh) by the weight of spray-dried secondary particles on the electrode, i.e., 79% of the total mass loading. Here, the electrodes are in principle cycled at 1 C under CCCV mode on lithiation and CC mode on delithiation. Capacity profiles of DOD100% are defined such that the reference electrode reaches a current limit of 0.01 C in the CV domain at 10 mV. Referring to the DOD100% capacity profiles over cycles, DOD is controlled by capacity-cutoff from 70~100% as shown in Fig. 4a and Supplementary Fig. 13a. For all experiments in this study, the first two cycles are carried out under DOD100% at 0.1 and 0.2 C to fully amorphize pc-SiNP, followed by subsequent DOD-controlled conditions at 1 C. Slower cycles at 0.02–0.1 C under DOD100% are inserted every 20 cycles regardless of the past DOD history, so as to capture structural characteristics through the whole voltage window.

**Accuracy of cycler.** The 2032-type coin cells are cycled in a commercial closed-system cycler (TOYO system, TOSCAT-3100 series). The internal temperature of the cycler is maintained at ~23 °C (±1 °C) during the measurements. The internal system is set to use a current acquisition pitch of ~1 s. As 50 identical channels are used in this study, the instrument can measure the current with an accuracy of ±0.0167% (167 p.p.m.) in a range of 2–10 mA. To confirm the reproducibility of our electrodes, six identically prepared control electrodes are cycled, and the difference in mass loading on Cu foil is kept within ~5%. The average SD of CE over 107 cycles is ±~0.07% (Supplementary Fig. 2). As the change in CE over the repetitive phase transformations is as much as ~2% over 150 cycles, these changes in CE can be detected due to our instrumental set-up and electrode reproducibility. Nevertheless, instruments an order higher in accuracy[34–36] are desirable, which we will pursue in the future.

**Reference electrochemistry.** The notation of Li–Si and Li–Gr processes follows a previous work[25]. Si#d-X/Si#c-X and Gr#d-X/Gr#c-X are Li–Si and Li–Gr processes; and #d-X and #c-X denote the X$^{th}$ discharge ('#d', lithiation) and charge ('#c', delithiation) processes in the half-cells, respectively. Since type-A electrodes include a portion of Gr, bare Gr-based electrodes are separately prepared to examine the background electrochemical processes. Three distinct processes are observed in them on both discharge and charge, with a capacity of ~300 mAh g$^{-1}$: dQ/dV peaks at 200 mV (Gr#d-1, 84 mAh g$^{-1}$ ≡ LiC$_{27}$), 110 mV (Gr#d-2, 171 mAh g$^{-1}$ ≡ LiC$_{13}$), and 80 mV (Gr#d-3, 300 mAh g$^{-1}$ ≡ LiC$_{7.4}$) on lithiation; and at 90 mV (Gr#c-1, 171 mAh g$^{-1}$ ≡ LiC$_{13}$), 140 mV (Gr#c-2, 80 mAh g$^{-1}$ ≡ LiC$_{28}$), and 230 mV (Gr#c-3) on delithiation. These dQ/dV processes are in good agreement with previous reports[42,43]. Figures 3d–g show that the Li–Gr processes can be separated from the Li–Si ones. The CE originating from the Li–Gr processes is above 99.5% after five cycles (Supplementary Fig. 24) and saturates to ~99.9% under DOD70–100%.

For type-A and -B electrodes, the process during the 1$^{st}$ lithiation is dominated by a sharp peak at 100 mV (Si#d-1, c-Si → c-Li$_{3.75(+\delta)}$Si; gradual lithiation of the c-Si lattice into a-Li$_x$Si, with further transformations into c-Li$_{3.75}$Si and c-Li$_{3.75+\delta}$Si)[25–27]. On delithiation, which is initiated by a rather flat process up to 300 mV (Si#c-1, c-Li$_{3.75(+\delta)}$Si → c-Li$_{3.75(-\delta)}$Si), the characteristic plateau at ~430 mV is dominant (Si#c-3, c-Li$_{3.75(-\delta)}$Si → a-Li$_{<\sim1.1}$Si; a signature of conversion of c-Li$_{3.75(-\delta)}$Si into a Li-substituted amorphous phase)[25]. A small 300 mV peak (Si#c-2, a-Li$_{\sim3.5-3.75}$Si→a-Li$_{\sim2.0}$Si) can be seen, which originates from residual a-Li$_x$Si at the end of lithiation[25]. In the following cycles, at least three distinct processes are observed at 300 mV (Si#d-2, a-Si → a-Li$_{\sim2.0}$Si), 100 mV (Si#d-3, a-Li$_{\sim2.0}$Si → a-Li$_{\sim3.5-3.75}$Si), and 50 mV (Si#d-4, a-Li$_{3.75}$Si → c-Li$_{3.75(+\delta)}$Si). Note that the process at 30 mV in the previous work[25] (Si#d-5, c-Li$_{3.75}$Si → c-Li$_{3.75+\delta}$Si; $\delta = \sim0.2$–0.3) is probably overshadowed by signals from the other components and merged in the Si#d-4 process in this study. The processes on delithiation are almost identical to the 1$^{st}$ cycle. These electrochemical Li–Si processes and the corresponding notations are summarized in Supplementary Table 5. At the probing points, type-A and -B electrodes are cycled at slower current rates (0.02–0.1 C) under DOD100% regardless of the DOD cycling history. The capacity retention rate at these points is defined with respect to the 1$^{st}$ reversible (delithiation) capacity. As shown in Supplementary Fig. 1, the retention rate at the probing points decreases suddenly by ~7.3% from DOD90% to 100% for both type-A and -B electrodes, while the difference between DOD90% and 80% is only ~1%. This observation is well known[7,26,30] to indicate that the a-Li$_x$Si → c-Li$_{3.75(+\delta)}$Si phase transformation has a greater impact on the degradation, compared with incremental amorphous Li$_x$Si volume changes under DOD80–90% cycling protocols ($x$~3.0–3.56). The difference in the retention rate between DOD100% and the other DOD controls gradually increases as cycling proceeds by up to ~7–8% around the 65$^{th}$ cycle, and then decreases to ~3–4% after the 107$^{th}$ cycle. Regarding a shift of the Li–Si processes from asymmetric to symmetric (Fig. 3h), it persists for the current rates from 0.02 to 1 C (Supplementary Fig. 5–7). The Coulombic irreversibility parameters display similar trends in type-A and -B electrodes (Fig. 4 and Supplementary Fig. 13). This leads to three conclusions: (i) the trend can be observed in systems that contain a portion of Gr in the electrodes, (ii) the reversibility characteristics in type-A electrodes is not due to Si capacity decay with a concurrent increase of Li–Gr processes that have inherently higher CE (>~99.5% after the 5$^{th}$ cycle, Supplementary Fig. 24), and (iii) these characteristics are not due to changes in the relative contributions of Li–Si and Li–Gr processes to the capacity under kinetics cycling conditions. Thus, importantly, the irreversible behaviors seen in Fig. 4 and Supplementary Fig. 13 can be generalized to other Si-rich electrode systems.

Regarding the Coulombic irreversible behaviors in Fig. 4, one issue requiring careful consideration is the difference in Si exposure time to the electrolyte for different DOD controls at CV on lithiation. This is because of the nature of CCCV cycling at higher current rates, in which the potential reaches the CV domain in the early stage, and therefore the CV duration can dominate the entire CCCV stage. Consequently, the CV duration can be much longer under DOD100% compared with the other DODs (Supplementary Fig. 11a, b and 12a, b for type-A and -B electrodes, respectively). Therefore, separate cycling experiments using 0.1 C throughout are also conducted under the same DOD controls, such that the CCCV duration can be controlled to be nearly proportional to DOD% at this rate (Supplementary Fig. 11c, d and 12c, d). The result shows that the CE profiles are very similar between 0.1 and 1 C (Supplementary Fig. 11e–g and 12e–g, respectively). These results suggest that the nonlinear CE transition between DOD70–90% and DOD100% is mainly triggered by repeated c-Li$_{3.75(+\delta)}$Si formation/decompositions, rather than by variation in Si exposure time to the electrolyte or incremental volume changes.

**Li-metal resistance.** Electrochemical impedance spectroscopy (EIS) is conducted for fully lithiated coin half-cells and symmetric cells for different DODs. The frequency is swept from 1 MHz to 0.1 Hz with a fluctuating voltage of ±5 mV. The cell is cycled at 1 C under different DOD controls for given number of cycles. Then,

the current rate is switched to 0.05 C on lithiation, and the potential is held at 10 mV for at least 24 h to stabilize the metastable c-$Li_{3.7(+\delta)}$Si. The semicircle at mid-range frequencies (10–10,000 Hz) in the EIS data for half-cells (Supplementary Fig. 8) becomes much smaller when measured using symmetric cells made of identical anodes (Supplementary Fig. 9). This raises a possibility that the increased Li-metal resistance in half-cells may limit the electrochemical Li–Si processes, which eventually eliminates Si#c-3. To explore this, a coin half-cell with type-A electrode is cycled 107 times under DOD100%. Afterwards, the cell is disassembled and then reassembled with fresh Li-metal, separator, and electrolyte in Ar-purged glovebox (Supplementary Fig. 10a). The reassembled cell is then cycled several times at 0.02 and 0.05 C under DOD100%. Supplementary Fig. 10b shows that Si#c-3 is still absent in the d$Q$/d$V$ profile. This indicates that the elimination of Si#c-3 or the shift from asymmetric to symmetric Li–Si processes is not due to kinetics (e.g., the resistance increase in Li-metal over cycles nor current rates), but to alteration of energetics in the Li–Si processes.

**TEM.** The electrodes are characterized using SEM by slicing with a Ga focused ion beam (FIB) at 5 keV acceleration (Helios Nanolab 450F1, FEI). Then, ex situ TEM analyses are carried out using a double-Cs-corrected Titan Cubed microscope (FEI) at 300 kV with a Quantum 966 energy filter (Gatan Inc.), and a probe Cs-corrected Titan 80-200 microscope (FEI) at 200 kV with a Super-X EDS detector. To avoid sample contamination and reaction upon air exposure, a vacuum transfer TEM holder (Model 648, Gatan Inc.) and transfer vessel for FIB (hand-made) are used. All samples are moved from the FIB transfer vessel to the vacuum transfer TEM holder in Ar-purged glovebox. Coin half-cells with type-A electrodes are cycled at 1 C over 107 cycles under different DOD controls. To fully delithiate the electrodes, the half-cell potential is held at 1.5 V for at least 24 h, until the current is less than 0.001 C. To avoid potential structural differences with respect to electrode positions, the analyzed section is always chosen at between 0 and 5 μm from the surface. The amorphized SiNP after the first two cycles is shown in Supplementary Video 1 and 2. The structure consists of extremely fine pores (~3.5 nm with SD ($\sigma$) ~0.8 nm) and quantum-size frames (~1.8 nm with $\sigma$~0.5 nm), which is segmented by 10-nm-thick chunky stripes. The amorphized structure could sustain the sphere (Fig. 5c and Supplementary Fig. 14). The porous delithiated a-SiNP condenses into bulk form at the $3^{rd}$ full lithiation at 10 mV. Thus, the structure keeps "breathing" during cycling, and the frames that appear in the delithiated structures may work as building blocks to gradually form the complex structure shown in Figs. 5d–g. Origin of the formation of the porous structure is to be presented elsewhere. Supplementary Fig. 15 shows the particle size and SD of the delithiated porous Si frame (from 50 to 100 randomly picked samples) under DOD80–100% after the $107^{th}$ cycle. They have Si feature size ($d$) ~6.4, 6.3, and 4.8 nm with $\sigma$ ~3.0, 2.7, and 1.0 nm, respectively. This clearly shows that the repeating c-$Li_{3.75(+\delta)}$Si formation/decomposition can accelerate reducing the Si size and sharpen its distribution. To observe the fully lithiated electrodes, the cell is cycled down to 10 mV at 0.05 C and held there until the current decays to less than 0.001 C. The cell is then disassembled in Ar-purged glovebox, and the electrode is promptly washed with dimethyl carbonate (DMC) for 5 min and then dried under vacuum for 30 min. The electrodes are then scraped onto a lacey-carbon TEM grid (Sigma-Aldrich). The grid is transferred to the TEM holder using an in-house airtight transfer vessel without exposure to the ambient air, followed by prompt TEM measurement due to the metastable nature of c-$Li_{3.75(+\delta)}$Si.

**XPS.** For ex situ XPS measurements (PHI Quantera-II), the core-level spectra are measured using Al Kα as the excitation source (1486.6 eV) at an acceleration voltage of 1 kV. The atomic concentrations are determined and curve fitting is carried out after Shirley background subtraction. All spectra are referenced to the C 1 s peak at 284.8 eV. Coin cells cycled under different DOD controls at 1 C are disassembled in Ar-filled glovebox and washed with DMC for 5 min, followed by 30 min drying under vacuum. Subsequently, the electrodes are loaded into an in-house airtight vessel and transferred to the instrument without exposure to the ambient air. Spectra are recorded for the electrodes before cycling, after the amorphization of c-Si, and at probing points during 107 cycles for DOD80%, 90%, and 100%. The bare electrode is soaked in the electrolyte prior to the DMC washing to observe signal from residual $LiPF_6$. Each electrode is analyzed after sputtering with Ar ions for different amounts of time (0–5 min) at the rate of ~6 nm/min to remove potential contamination and oxidation on the surface. The $SiO_x/Li_xSiO_y$ peak always accompanies the Si signal, both of which only start after 1 min of etching. This observation indicates that $SiO_x/Li_xSiO_y$ is located in close proximity to Si. As shown in Supplementary Fig. 20, the Li 1s, F 1s, and Si 2p depth profiles indicate that the majority of F-related composites are made of LiF with fractions of residual $LiPF_6$ in the surface region, or (Li)$P_xO_zF_z$. Note that the amount of P in the electrodes is less than 0.5 at% for all samples, regardless of the number of cycles and etching depth. LiF may originate mostly from defluorination of FEC[45,46]. Li–X (X = $O_{0.5}$, O, OH) species are also found in the profiles, although it is difficult to clearly distinguish between oxide and non-oxide components in the Li-1s spectra.

**XRD.** For ex situ XRD measurements, the coin half-cells are cycled at 1 C under different DOD controls until the probing points. Subsequently, the electrodes are slowly lithiated at 0.05 C and the potential is maintained at 10 mV for at least 24 h

until the current decays to less than 0.001 C to stabilize the metastable c-$Li_{3.75(+\delta)}$Si. Ogata et al.[25] showed that relaxation of the metastable phase becomes sluggish when cycled in this manner. After the cycling, the coin cells are immediately disassembled in Ar-filled glovebox, sealed with airtight Kapton tape, and immediately transferred to the XRD instrument (Bruker, D8 Advance). Conventional XRD measurements are performed using Cu Kα radiation (1.54 Å). Each spectrum is acquired in the range of 5–80° (2$\theta$) for ~50 min. The c-$Li_{3.75(+\delta)}$Si (332) reflection peak is fit by the Voigt function using a free software (Fytik) to determine its FWHM. The error in the FWHM is estimated to be 0.05°, considering the data acquisition pitch of the instrument.

**$^7$Li solid-state NMR spectroscopy.** MAS $^7$Li solid-state (ss-)NMR experiments are performed on Bruker Avance III consoles, $^1$H Larmor frequency of 600.13 MHz (14.1 T). Commercial Bruker double-resonance 2.5-mm MAS probes that allow spinning frequencies up to 35 kHz are used for all experiments. $^7$Li MAS NMR spectra (233.2 MHz) are acquired ex situ at a spinning rate of 15 kHz with $\pi$/2-(one-pulse) measurements with a 2.0 s last-delay duration over 64 scans. After the coin half-cells reach the probing points at 1 C under different DOD controls, the cells are cycled at 0.05 C until reaching the target potential and held there for at 24 h, until the current decays to less than 0.001 C. The cell is then immediately disassembled in an Ar-filled glovebox, dried for at least 30 min under vacuum, and packed in the rotor for the NMR measurements. All the $^7$Li ss-NMR chemical shifts are referenced to 1 M LiCl (sol.) at 0 p.p.m. as an external reference. The correlation of $^7$Li ss-NMR chemical shifts with Li–Si environments and with the electrochemical processes is summarized in Supplementary Table 5. Spectra are recorded at 300, 150, 80, and 10 mV on lithiation and 150, 250, and 550 mV, and 1.5 V on delithiation at the probing points for DOD100% and 90% over 107 cycles. On the basis of the previous studies[25], the $^7$Li resonances are linked to Li–Si local environments as follows: 10–0 p.p.m. corresponds to larger Si clusters and extended Si networks; 25–10 p.p.m. to small Si clusters; 6–0 p.p.m. to isolated $Si^{4-}$ anions including c-$Li_{3.75}$Si; and 0 to −10 p.p.m. to overlithiated crystalline phase of c-$Li_{3.75+\delta}$Si.

**XAFS.** Ex situ XAFS at the Si K-edge is measured at BL-10 of Synchrotron Radiation (SR) Center at Ritsumeikan University. The photon beam energy delivered to the samples ranges from 1000 to 2500 eV with a resolution of 0.5 eV or less. 2032-Type coin half-cells are cycled at 1 C under the designated DOD controls to the target cycle number. To fully delithiate the electrode, the half-cell potential is held at 1.5 V for at least 24 h until the current decays to 0.001 C. The cells are then disassembled in an Ar-filled glovebox. The electrode is rinsed with DMC for 5 min, set on carbon-taped sample holders, loaded into an airtight vessel, and then transferred to the BL-10 chamber without exposure to ambient air. The vessel is immediately evacuated, and the samples are loaded into the measurement chamber with a vacuum level of $5 \times 10^{-8}$ Pa. Partial fluorescence yield mode is adopted to measure XAFS over the EXAFS range for the Si K-edge, which enables effective elimination of the P K-edge absorption signal by energy-selected fluorescence detection with a Si drift detector. Small amounts of residual P on the surface of the Si anode could not be completely removed even after rinsing, which agrees with the XPS results that show <0.5 at% P in all the samples at all etching durations. The total electron yield is also simultaneously measured, and P is detected in the EXAFS region of Si. For XANES (Supplementary Fig. 22a), after the initial amorphization, the $SiO_2$ peak shifts to a lower energy of 1843–1846 eV ($SiO_x$ and/or $Li_ySiO_z$), indicative of native oxides on Si forming Li silicate or being further oxidized. As cycling progresses, this absorption increases for all DODs, being more prominent at higher DOD100% (Supplementary Fig. 22a). Using open source analysis software (Athena), EXAFS (Supplementary Fig. 22b) data for delithiated amorphous-Si (a-Si) at the probing points are extracted from XAFS and Fourier-transformed, making them equivalent to RDF profiles. Tabulating the coordination number of a-Si involves a few uncertainties, such as statistical EXAFS fitting errors, sample preparation reproducibility, and assumptions made during data analysis for the physical structures surrounding the absorber[47]. Hence, here we indeed integrated the 2 Å Si–Si correlation peaks to index the Si–Si tetrahedral environments in the delithiated a-Si. Oxidation of the anodes in the ambient environments after cycling is probably minimized, as the amplitude of the 1 A coordination peaks in the Fourier-transformed profiles is lower than ~0.3–0.4. The error in $A_{(2Å \ Si–Si)}$, originating from sample reproducibility and handling issues, is not large enough to invalidate the overall trend observed in Figs. 8b, c (probably ~10% at most). In Fig. 8c, it should be noted that when the $A_{(2Å \ Si–Si)}$ over cycles starts to decrease at ~0.95, the same path is not followed for all DOD conditions. When $A_{(2Å \ Si–Si)}$ starts to decrease, higher CEs are observed instead for the same $A_{(2Å \ Si–Si)}$. This change is probably caused by the evolving morphology in a-Si, from being more entangled/agglomerated to sharper sub-5-nm structures and by the changes in Si interfacial property/energetics (Supplementary Fig. 20, 21, and 23).

**Numerical calculations.** Bulk amorphous $Li_xSi$ structures are constructed using a series of melting, quenching, and relaxation processes of thermodynamically stable crystalline $Li_xSi$ structures ($x$ = 2.33, 3.25, and 3.75)[48]. The structures are obtained using ab initio calculations based on DFT implemented in the Vienna ab initio simulation package[49], in which the generalized gradient approximation suggested

by Perdew, Burke, and Ernzerhof [50] is adopted for the exchange-correlation functional, and the projector augmented wave method[50] is used for the atomic quasi-potentials of all elements. To ensure amorphism of the generated structures, a sufficiently large number of atoms are included in the supercells: 120, 136, and 152 atoms for $x = 2.33$, 3.25, and 3.75, respectively. By using ab initio molecular dynamics simulation, the c-Li$_x$Si structures are melted at 4000 K for 5 ps with 1 fs time steps, and then quenched at 300 K by assuming the canonical ensemble based on the Nosé algorithm. Here, k-points of $1 \times 1 \times 1$ with $\Gamma$ symmetry-point-centered sampling and a cutoff energy of 300 eV for the plane-wave basis are used. Subsequently, full structural relaxation of the quenched structures by DFT leads to the final bulk a-Li$_x$Si structures, in which we used an atomic force tolerance of 0.02 eV/Å, electronic energy tolerance of $10^{-6}$ eV, energy cutoff of 500 eV, and $\Gamma$-centered k-point sampling of $2 \times 4 \times 2$, $2 \times 2 \times 4$, and $3 \times 3 \times 2$ for $x = 2.33$, 3.25, and 3.75, respectively. The density of the amorphous structures is determined by thermodynamic evolution of crystalline structures with well-defined density. The initial spherical a-Li$_x$Si nanoclusters are created with the same 84 Si atoms for all Li fractions, by using bulk amorphous Li$_x$Si structures previously obtained by DFT and preserving the relative atomic coordinates. The diameters of a-Li$_x$Si are 20.83, 22.41, and 23.42 Å and the total numbers of atoms contained in the clusters are 280, 357, and 399 for $x = 2.33$, 3.25, and 3.75, respectively. In addition, slightly larger amorphous bulk structures are regenerated with the amorphous bulk structures obtained by DFT. The final spherical a-Li$_x$Si clusters and bulk structures are obtained by performing a classical molecular dynamics simulation implemented in the Large-scale Atomic/Molecular Massively Parallel Simulator (Lammps) package[51] with the reactive force field (ReaxFF)[52], as shown in Supplementary Fig. 23. The structures are relaxed at 300 K under a Nosé-Hoover thermostat for 1 ns with 1 fs time steps in a canonical (NVT) ensemble. The FE for a given Li fraction $x$ is calculated as:

$$\mathrm{FE}(x) = E_{\mathrm{Li}_x\mathrm{Si}} - xE_{\mathrm{Li}} - E_{\mathrm{Si}} \tag{1}$$

where $E_{\mathrm{Li}_x\mathrm{Si}}$ is the total energy of a Li$_x$Si structure divided by the number of Si atoms, and $E_{\mathrm{Li}}$ and $E_{\mathrm{Si}}$ are energies per atom in the body-centered cubic (bcc) structure of Li and diamond structure of Si, respectively. The surface energy is given by:

$$\sigma = \frac{1}{A}\left(E_{\mathrm{sphere}} - E_{\mathrm{bulk}}\right) \tag{2}$$

where $E_{\mathrm{sphere}}$ is the total energy of the Li$_x$Si spherical cluster, $E_{\mathrm{bulk}}$ is the bulk energy of Li$_x$Si with the same number of Si atoms as in the spherical cluster with the corresponding $x$ value, and $A$ is the surface area of the spherical cluster. The key question to be addressed by simulation is: why is $+\delta$ absent in c-Li$_{3.75(+\delta)}$Si at the probing points in the symmetric regime despite the presence of c-Li$_{3.75}$Si. The formation of c-Li$_{3.75+\delta}$Si is energetically favorable in the event of c-Li$_{3.75}$Si formation, owing to the lower-energy cost of inserting Li atoms into c-Li$_{3.75}$Si nuclei than that of breaking residual Si–Si bonds[25]. Attributing the absence of $+\delta$ to the increasing resistance from accumulated SEI and degraded electric conduction network over Si expansion/contraction might not suffice, since $+\delta$ is absent under the potentiostatic quasi-thermodynamic cycling conditions (Supplementary Figs 6 and 7). In the nanocluster, the gradient for the decrease in formation energies ($\xi$), which measures the driving force for lithiation, is significantly suppressed at around $x = 3.25$ ($\xi = -0.177$ and $-0.028$ eV for $x = 2.33$–3.25 and 3.25–3.75, respectively), whereas that in the bulk barely changes ($\xi = -0.23$ and $-0.297$ eV for the same intervals of $x$). This sudden decrease in the driving force is partly attributed to an increased surface energy in the structures. Hence, in such nanoclusters, there is less momentum to reach $x = 3.75$ ($+\delta$) due to the decreased $\xi$ and increased surface energy, which may result in more uniform lithiation by breaking residual Si–Si bonds rather than locally inserting extra Li atoms into c-Li$_{3.75}$Si nuclei. This may at least be partly responsible for the absence of $+\delta$ in c-Li$_{3.75(+\delta)}$Si in the symmetric regime.

**Data availability**. The authors declare that all data supporting the findings of this study are available within this article, its Supplementary Information files, or are available from the corresponding author upon reasonable request.

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

## Acknowledgements

We acknowledge discussion on XAFS analyses with Toshiaki Ohta, Koji Nakanishi, and Toyonari Yaji, and useful advice from Makoto Ue on a manuscript scheme.

## Author contributions

K.O., S.J., and D.-S.K. designed the experiments. K.O. and T.K. designed and synthesized the active materials. K.O. and S.S. prepared all the electrodes and acquired all the electrochemical data. D.-S.K. and H.-G.K. acquired the TEM images. K.I. and Y.K. acquired XAFS data. I.S.J. and J.H.K. acquired and processed the NMR data. K.I. and Y.K. acquired and processed XAFS data. J.-H.K. acquired XRD data. Y.-H.C. conducted DFT calculations. H.P., J.J., and M.K. synthesized electrolytes. Y.S.K. acquired XPS data. M.K., J.J., H.P., W.C., S.G.D., Y.H., K.U., and S.H. provided insights on the experiments. All authors wrote the article.

## Additional information

**Competing interests:** The authors declare no competing financial interests.

K. Ogata [1,2], S. Jeon [1], D.-S. Ko[1], I.S. Jung[1], J.H. Kim [1], K. Ito[3], Y. Kubo [3], K. Takei[1], S. Saito[2], Y.-H. Cho[1], H. Park[1], J. Jang[1], H.-G. Kim[1], J.-H. Kim[1], Y.S. Kim[1], W. Choi[1], M. Koh[1], K. Uosaki [3], S.G. Doo[1], Y. Hwang[1] & S. Han[1]

[1]Samsung Advanced Institute of Technology, Samsung Electronics, Samsung-ro 130, Suwon, Gyeonggi-do 16678, Korea. [2]Samsung Research Institute of Japan, Samsung Electronics, 2-1-11, Senba-nishi, Mino-shi, Osaka-fu 562-0036, Japan. [3]C4GR-GREEN, National Institute for Materials Science, 1-1 Namiki, Tsukuba, Ibaraki 305-0044, Japan. K. Ogata, S. Jeon and D.-S. Ko contributed equally to this work.

