## [Peer Review File · Nature Communications]

Reviewers' comments:

Reviewer #1 (Remarks to the Author):

This paper demonstrates the capability to determine the impact of Li-Si phase transformations (particularly the formation of $c\text{-Li}_{3.75+\delta}\text{Si}$) on the reversibility and structural changes in nanostructured Si-based anodes by controlling the lithiation states in coin half-cells, i.e., the depth of discharge (DOD). However, many similar studies have been reported in the literature [e.g., Journal of Power Sources 189 (2009) 34–39, Journal of Power Sources 189 (2009) 1132–1140, Journal of Power Sources 161 (2006) 1254–1259, J. Phys. Chem. C (2009), 113, 11390–11398, Chem. Mater. (2012), 24, 1107–1115]. While this work confirmed that the control of DOD leads to the structure and phase changes in nanostructured Si-based anodes, only these changes could not be sufficient for explaining the variations of Coulombic efficiency. It is more critical to clarify how the resulting volume change, pulverization as well as the degradation of SEIs critically influence the electrochemical performance. However, the results presented (e.g., ex situ TEM) are not clear enough to provide new mechanistic insights that are important enough to the field. Hence, this work does not contain sufficient novel results to warrant its publication in Nature Communications.

Additionally, the following issues should be addressed before next submission.

- It is difficult to read this paper. The authors put too many details into the main text. It is hard to separate the novel findings and insights from the well-known facts. Also, there are many uncommon acronyms used, such as FP, RP, VE, etc. Many sentences are overly long.
- Do the authors normalize the NMR data, given the ex-situ NMR reported?
- The scan range of NMR should be increased.

Reviewer #2 (Remarks to the Author):

Review of "In situ tunable links between key Li-Si structural transformations and Coulombic reversibility" by K. Ogata et al.

In this study, the authors report an extensive investigation into the links between depth of discharge, Coulombic efficiency, and the formation of the $\text{Li}_{15}\text{Si}_4$ phase during cycling of silicon anodes for Li-ion batteries. A significant electrochemical effort, complemented by ex situ TEM, XRD, NMR, XAFS, and DFT, is pursued. The research shows that there is indeed a link between CE and the formation of the $\text{Li}_{15}\text{Si}_4$ phase (among other conclusions), which may be important for designing electrodes and full cells for application of Si in commercial cells.

This study presents an enormous amount of data, and the work is generally well done and important for progress of Si anodes. However, I expect that this paper will be of most interest to experts in the field of alloying anodes for Li-ion batteries, and of less interest to general materials/electrochemical scientists, as required by Nature Communications. The highly specialized nature of the manuscript is exemplified by the considerable use of acronyms, shortened sample names, and abbreviated labels which makes the manuscript difficult to follow in some points, even for experts. I understand the authors' dilemma – the data are extensive and complex, and they have done a commendable job of organizing and presenting it. However, the very nature of the findings in this paper perhaps point to a more specialized and non-"communications" journal in which the findings can be presented in full. Based on these comments, I think it should be up to the editors' discretion to determine whether this paper is suitable for Nature Communications.

A few additional suggestions:

-A minor note: in the introduction, the authors state that previous work has shown "large asymmetric VE owing to different Li-diffusivity through different c-Si facets." While there was some initial confusion regarding this phenomenon, the consensus of the field is that this happens due to anisotropic reaction rates at different facets, not anisotropic diffusivity within the crystal. This should be corrected in the introduction.

-In fig. 2a, why does the DOD100% data have error bars, but the other data do not?

-The authors state that the $\text{Li}_{3.75}\text{Si}$ phase is clearly indexed in Fig. 4, but the SAED patterns are not labeled, and they are very small and difficult to discern.

-In Fig. 4g, the colored points are not labeled with 100%, 90%, 80% DOD.

-I appreciate the effort in Fig. 7 to simplify the conclusions with a schematic. However, this schematic is quite complex – there are no a, b, c labeled sections that are referred to in the caption, and it is difficult to know where to even start examining this figure. Also, are the colors in the pie charts labeled anywhere?

Our point-by-point responses to reviewer's comments

We have three responses both to Reviewer #1 and #2, as following,

For **Reviewer #1**,

Response #1-3 (> Reviewer #1)

For **Reviewer #2**,

Response #4-6 (> Reviewer #2)

These responses are shown in the followings in the order of a chronological order. Note that the reviewers' comments are **highlighted by blue colour**.

Dear Reviewer #1

*******Reviewer #1 comment starts*******

Reviewer #1 (Remarks to the Author):

This paper demonstrates the capability to determine the impact of Li–Si phase transformations (particularly the formation of $c\text{-Li}_{\{3.75+\delta\}}\text{Si}$) on the reversibility and structural changes in nanostructured Si-based anodes by controlling the lithiation states in coin half-cells, i.e., the depth of discharge (DOD). However, many similar studies have been reported in the literature [e.g., Journal of Power Sources 189 (2009) 34–39, Journal of Power Sources 189 (2009) 1132–1140, Journal of Power Sources 161 (2006) 1254–1259, J. Phys. Chem. C (2009), 113, 11390–11398, Chem. Mater. (2012), 24, 1107–1115]. While this work confirmed that the control of DOD leads to the structure and phase changes in nanostructured Si-based anodes, only these changes could not be sufficient for explaining the variations of Coulombic efficiency. It is more critical to clarify how the resulting volume change, pulverization as well as the degradation of SEIs critically influence the electrochemical performance. However, the results presented (e.g., ex situ TEM) are not clear enough to provide new mechanistic insights that are important enough to the field. Hence, this work does not contain sufficient novel results to warrant its publication in Nature Communications.

Additionally, the following issues should be addressed before next submission.

- It is difficult to read this paper. The authors put too many details into the main text. It is hard to separate the novel findings and insights from the well-known facts. Also, there are many uncommon acronyms used, such as FP, RP, VE, etc. Many sentences are overly long.
- Do the authors normalize the NMR data, given the ex-situ NMR reported?
- The scan range of NMR should be increased.

*******Reviewer #1 comment ends*******

Thank you very much for sparing your precious time amongst your busy schedule for reviewing our manuscript. Our replies to Reviewer #1 are summarised in the followings, consisting of 3 responses labelled **Response #1-3 (> Reviewer #1)**

Response #1 (> Reviewer #1)

Reviewer #1 comments "However, many similar studies have been reported in the literature", at least listing five literatures as followings,

(**Literature #1**). Candace K. et al., "Structural and electrochemical study of the reaction of lithium with silicon nanowires" <http://www.sciencedirect.com/science/article/pii/S0378775308023616>

(**Literature #2**). Candace K. et al., "Surface chemistry and morphology of the solid electrolyte interphase on silicon nanowire lithium-ion battery anodes"

<http://www.sciencedirect.com/science/article/pii/S0378775309000524>

(**Literature #3**). N.S. Choi. et al., "Effect of fluoroethylene carbonate additive on interfacial properties of silicon thin-film electrode" <http://www.sciencedirect.com/science/article/pii/S0378775306010408>

(**Literature #4**). R. Ruffo et al., "Impedance Analysis of Silicon Nanowire Lithium Ion Battery Anodes" <http://pubs.acs.org/doi/full/10.1021/jp901594g>

(**Literature #5**). B. Philippe et al., "Nanosilicon Electrodes for Lithium-Ion Batteries: Interfacial Mechanisms Studied by Hard and Soft X-ray Photoelectron Spectroscopy"

<http://pubs.acs.org/doi/abs/10.1021/cm2034195>

We notice the elegance of these works. However, the works are clearly irrelevant to the central point in our work. Prior to explaining the irrelevancy, we need to rephrase our "**Key point**" with relevant backgrounds.

Key point:

"Quantifying and qualifying key correlations between Coulombic efficiency (CE) evolutions and the hysteretic amorphous–crystalline (*a-c*) Li–Si phase transformations"

Key point#1: Iterating the hysteretic $c\text{-Li}_{3.75(+\delta)}\text{Si}$ (de)formation, usually featured as capacity degradation factors, can upheave CE in the most efficient way among the given Li–Si reaction paths, which is quantitatively distinguished from iterating the *amorphous* Li–Si reaction paths. For the first time, a positive aspect of $c\text{-Li}_{3.75(+\delta)}\text{Si}$ is highlighted.

Key point#2: The iterative $c\text{-Li}_{3.75(+\delta)}\text{Si}$ (de)formation accelerates a regime shift of Li–Si processes from an asymmetric to a symmetric reaction sequence, which is numerically associated with the CE behaviours in **Key point#1**. The iteration also accelerates a regime shift of Li–Si structural characteristics from a bulk to a surface dominated system, being probed by complimentary atomistic probing methodologies as well as DFT calculations.

Key point#3:

The regime shifts of electrochemical Li–Si processes and Li–Si structural characteristics in **Key point#2** are mutually correlated, which is further synchronised with the evolving CE behaviours in **Key point#1**.

Thus, our manuscript is all about CE. We primarily focus on quantitatively and qualitatively resolving the CE alterations by the different Li–Si electrochemical processes. More specifically, it is key to do so, by separating the alteration by the *a-c* phase transformations and by incremental *amorphous* Li–Si volume changes. Details of **Key point#1-3** are correlated with our new electrochemical and mechanistic findings, which are summarised in newly added **Supplementary Table 4** and **Table 5** in **Supplementary information** in the modified manuscript (shown in page 14 and 15 in this letter).

On the other hand, the literatures listed by Reviewer #1 basically argue how capacity retention and/or Li–Si structure/interface are changed by switching presence/absence of the *a–c* phase transformation, or vice versa. Hence, the literatures clearly have different objectives from ours.

Importantly, to resolve such CE behaviours in the perspective of **Key point#1-3**, we had to satisfy at least two prerequisites as summarised in the following “**Prerequisite #1–2**”.

Prerequisite #1

CE error due to instrumental precision and electrode reproducibility needs to be constrained small enough to argue potential CE alterations by the different Li–Si structural changes. CE precision in the light of our target needs to be less than ~0.1% at least.

Prerequisite #2

The anodes need to be designed so that almost all the *a*-Li_xSi components transform into *c*-Li_{x3.75(+δ)}Si abruptly at the end of the 1st lithiation, in which the empirical capacity needs to be very close to the theoretical capacity of Si (3579 mAh/g) so as to clearly define depth of discharge (DOD) over cycling.

For **Prerequisite #1**, we used a closed-box-type cycler (TOYO system, TOSCAT-3100 series) that sustains a constant temperature (23 °C) and reasonably calibrated the current acquisition precision (~167 ppm) (written in **Methods** > *Interpretation of cycler’s accuracy*). Further, to secure reproducibility of the electrodes, active materials is fabricated with spray-dry processes which well defines active material parameters (**Supplementary Table 1** and **Methods** > *Baseline active materials*). These efforts led to a CE accuracy of ±~0.07%.

For **Prerequisite #2**, many of previous works on the anodes do not even reference an empirical capacity to the theoretical one (~3579 mAh/g); typically, an empirical capacity is much lower than 3579 mAh/g due to kinetics limit on lithiation or poor electrode designs. In such anodes, it is hard to well define depth of discharge X% (DOD X% in a half-cell) in proportion to x in Li_xSi, and accordingly undergo the abrupt *a–c* transformation near DOD100%. Consequently, well-defined DOD controls in the following cycles get vague. Also, in such anodes, it is also difficult to interpret whether the retention in such anodes originates from capacity sustenance or from a balance between active material loss and gradual activation of kinetically unused capacities. Hence, in this work, we design the electrodes such that the empirical reversible capacity becomes very close to the theoretical one by ~99 % accuracy (**Supplementary Table. 1**), which is defined as DOD100%. Further, almost all the *a*-Li_xSi components abruptly convert into *c*-Li_{3.75(+δ)}Si between DOD95 and 100% in the initial cycles (**Methods** > *Baseline active materials*). This requires industry-level controls over active material synthesis and electrode fabrications.

However, most of previous anode works, including the listed literatures, do not even consider “**Prerequisite #1–2**”. Without the considerations, it is very difficult to resolve subtle CE alterations over longer cycling under realistic cycling conditions. These are/were certainly summarised in **Introduction** and **Methods** part in the manuscript.

For the irrelevancy of the listed literatures by Reviewer #1, we explain each by each.

For (**Literature #1**), Li-Si structural changes and the corresponding capacity retention are investigated for dozens of cycles by modulating operation potentials. This work discusses a correlation between capacity fading and the *a-c* transformation for a short-term. Also, the different structural evolutions depending on the cut-off potentials are discussed. However, CE is hardly discussed in the literature. Further, the literature is within/near the scope of the early pioneering works by Obrovac and Dahn, and recent works by Key and Ogata as shown in the following, all of which are certainly cited in our manuscript. In all these literatures, there is no consideration of “**Prerequisite #1-2**” in the first place. This shows that the literatures’ primary objectives are clearly different from ours. Hence, we thin **Literature #1** is irrelevant to our works.

Our ref, Obrovac, M. N. & Krause, L. J. Reversible cycling of crystalline silicon powder. *Journal of the Electrochemical Society***154**, A103-A108, doi:Doi 10.1149/1.2402112 (2007).

<http://jes.ecsdl.org/content/154/2/A103.full>,

Our ref, Ogata, K. et al. Revealing lithium-silicide phase transformations in nano-structured silicon-based lithium ion batteries via in situ NMR spectroscopy. *Nat Commun*, **5**, (2014).

<https://www.nature.com/articles/ncomms4217>

Our ref, Key, B., Morcrette, M., Tarascon, J. M. & Grey, C. P. Pair Distribution Function Analysis and Solid State NMR Studies of Silicon Electrodes for Lithium Ion Batteries: Understanding the (De)lithiation Mechanisms. *Journal of the American Chemical Society***133**, 503-512, (2011).

<http://pubs.acs.org/doi/abs/10.1021/ja108085d>

Our ref, Hatchard, T. D. & Dahn, J. R. In situ XRD and electrochemical study of the reaction of lithium with amorphous silicon. *Journal of the Electrochemical Society***151**, A838-A842, (2004).

<http://jes.ecsdl.org/content/151/6/A838.full>,

Our ref, Obrovac, M. N. & Christensen, L. Structural changes in silicon anodes during lithium insertion/extraction. *Electrochemical and Solid State Letters***7**, A93-A96, (2004).

<http://esl.ecsdl.org/content/7/5/A93.full>,

For (**Literature #2 and #5**), there is no discussion on CE and it is simply irrelevant to our manuscript.

For (**Literature #3**), how electrolyte compositions affect electrode cyclability is discussed. However, our primary objective is to reveal correlations between CE and various Li-Si structural changes. Hence, varying electrolyte is out of the scope of our manuscript and irrelevant to our objectives. Input parameters to be controlled and output parameters to be probed, in this study, are summarised in newly added **Supplementary Table 3** in **Supplementary Information** (shown in page 13 in this letter).

For (**Literature #4**), this work provides detailed Si/SEI interfacial analysis by XPS and impedance analysis by EIS, controlling the *a-c* phase transformation by modulating operating voltage in coin half cells. This is an excellent work on the interfacial details, however, again, CE is hardly shown. Hence, the objective of this study is different from ours.

Based on those, the literatures listed by the reviewer have clearly different objectives from ours.

Previous Si anode works on CE are not numerous, but there are several recent works (cited in the 1st version of our manuscript) as followings,

Our ref, Bond, T. M., Burns, J. C., Stevens, D. A., Dahn, H. M. & Dahn, J. R. Improving Precision and Accuracy in Coulombic Efficiency Measurements of Li-Ion Batteries. Journal of The Electrochemical Society **160**, A521-A527, (2013). <http://jes.ecsdl.org/content/160/3/A521.abstract>

Our ref, Li, Y. et al. Growth of conformal graphene cages on micrometre-sized silicon particles as stable battery anodes. Nature Energy **1**, 15029, (2015). <https://www.nature.com/articles/nenergy201529>

Our ref, Ko, M. et al. Scalable synthesis of silicon-nanolayer-embedded graphite for high-energy lithium-ion batteries. Nature Energy **1**, 16113, (2017). <https://www.nature.com/articles/nenergy2016113>

Our ref, Jin, Y. et al. Self-healing SEI enables full-cell cycling of a silicon-majority anode with a coulombic efficiency exceeding 99.9%. Energy & Environmental Science, (2017).

<http://pubs.rsc.org/en/Content/ArticleLanding/2017/EE/C6EE02685K#!divAbstract>

The first work by Prof. Jeff Dahn mainly focuses on CE precision in terms of instrumental precision issues. The following three works tackle achieving higher CE by fabricating the nano-engineered composites. However, as far as we notice, there is little work that focuses on fundamental insights into CE correlations with various Li-Si structural changes in the light of “**Key point**” under “**Prerequisite #1–2**”.

Based on all of these, it is very hard for us to agree with the relevancy of the 5 listed literatures to our work. However, we admit that the lengthy/redundant writing style in the 1st version of our manuscript was just a burden for readers to certainly digest our essence. Therefore, now we significantly reduced the main text and more concisely rephrased the details. Further, to provide a clear overall experimental scheme for the enormous dataset, **Supplementary Table 3** (shown in page 13 in this letter) is added in **Supplementary Information**. Also, our electrochemical and mechanistic new findings are summarised in newly added **Supplementary Table 4** and **Table 5** in **Supplementary Information** (shown in page 14 and 15 in this letter).

Response #2 (>Reviewer #1)

Reviewer #1 comments “While this work confirmed that the control of DOD leads to the structure and phase changes in nanostructured Si-based anodes, only these changes could not be sufficient for explaining the variations of Coulombic efficiency. It is more critical to clarify how the resulting volume change, pulverization as well as the degradation of SEIs critically influence the electrochemical performance. However, the results presented (e.g., ex situ TEM) are not clear enough to provide new mechanistic insights that are important enough to the field.”

In the first place, we need to insist that consequence of arguing what Reviewer #1 claims insufficient, i.e. validity of the mechanism behind the CE alterations, does not affect validity of **Key point#1** and **#2**, anyway. Our key claim, mostly based on **Key point#1** and **#2**, can be established regardless of the validity in **Key point#3**.

In the second place, but more importantly, we complementarily interpret multiple analytical results to argue the mechanism behind the CE behaviours, including what Reviewer #1 raised (e.g. pulverisation, volume expansion, and SEI formation). To highlight our interpretations more clearly, we newly added **Supplementary Table 3-5** in *Supplementary Information* in the modified manuscript, which summarises our experimental scheme (**Supplementary Table 3**), the new electrochemical and mechanistic findings (**Supplementary Table 4** and **Table 5**). These tables are also shown in page 13–15. Importantly, the electrochemical and structural behaviours in the new findings are mutually correlated, with which the regime shifts and the CE behaviours are also synchronised. Thus, we interpret the mechanism behind the CE alterations. Although the mechanism can be further explored in more details, we believe that our findings and the comprehensive interpretations provide key proofs that are associated with the CE behaviours. Hence, it is not true that we correlate the CE alterations solely from a specific analytical result, e.g. TEM or Li–Si structural/phase transformations.

In the following, we specifically argue the comments by Reviewer #1. For the remarks, “**It is more critical to clarify how the resulting volume change, pulverization as well as the degradation of SEIs critically influence the electrochemical performance.**”, we considered the issues in the 1st version of our manuscript by the following analyses labelled by colour schemes in **Supplementary Table 3** (page 13 in this letter).

- *volume change: **Material analysis#3**,
- *pulverization: **Output#2, Material analysis#1, 2, 8**
- *SEIs: **Material analysis#1, 2, 3, 5, 6, 9**

Moreover, we discussed the following issues by the various probing methodologies as summarised in **Supplementary Table 5**.

- **a*-Si feature size changes over cycles: **Material analysis#1**
 - **c*-Li_xSi crystal size changes over cycles: **Material analysis#1, 4**
 - *Charge transfer resistance using symmetry cells over cycles: **Material analysis#5**
 - *Changes of Si local environments in *a/c*-Li_xSi over cycles: **Material analysis#7**
 - *Statistical population changes of Si-Si nearest tetrahedral correlation over cycles: **Material analysis#8**
 - *Surface energetics shift from bulk to the sub-5nm Si structures: **Material analysis#9**
 - *The regime shifts over cycles: **Material analysis#3,4,7,, Output #3**
 - *CE susceptibility to *c*-Li_{3.75(+δ)}Si presence/absence in each regime: **Material analysis#8, Output #1**
 - *Electrode thickness over 190 cycles with different DOD controls: **Material analysis#3**
 - *Extra capacity degradation due to iterative *c*-Li_{3.75(+δ)}Si (de)formation over cycles: **Output #2**
- etc,

These structural/interfacial characteristics are primarily altered by the accumulative iteration of c - $\text{Li}_{3.75(+\delta)}\text{Si}$ (de)formation, which is clearly distinguished from incremental *amorphous* Li–Si volume changes by the DOD controls. As summarised in **Supplementary Table 4**, the iteration also alters the electrochemical reaction pathways from the asymmetric to symmetric regimes as well as the CE behaviours. Complimentarily combined all these together, the regime shift in the structural/interfacial characteristics and in the consequent Li–Si reaction mechanism is clearly associated with the CE behaviours. These are explained in more details in *Discussion* in the modified manuscript.

This is how we interpret the mechanism behind the CE alterations. Based on these, it is hard for us to agree with the corresponding reviewer's comments.

Response #3 (>Reviewer #1)

Additionally, the following issues should be addressed before next submission.

- It is difficult to read this paper. The authors put too many details into the main text. It is hard to separate the novel findings and insights from the well-known facts. Also, there are many uncommon acronyms used, such as FP, RP, VE, etc. Many sentences are overly long.

We really appreciate your comments. We seriously take the issue, because this can distract readers' understandings on our main points. To improve clarity of the main text,

1. We have reduced the total volume and used much less abbreviations.
2. We rephrased unclear text and deleted redundant sentences so that readers can easily interpret our essence.
3. Detailed yet necessary text to support our main claim is moved back into **Methods**.

- Do the authors normalize the NMR data, given the ex-situ NMR reported?

Yes. The purpose of showing NMR spectra in this study is to qualitatively provide the NMR profile changes over cycling at the give potentials. Hence, we in principle normalise each NMR profile such that the maximum peak gets close to 1. The amount of an electrode loaded into 2.5 mm NMR rotor is basically fixed to half of $\Phi 12$ mm electrode. However, the NMR signal intensity differs depending on Li concentration at each potential, distribution of the powder in the rotor, etc.

- The scan range of NMR should be increased.

We took the spectra with 200 ~ -200 ppm scan range. Outside of this range, there is no useful information except spin side bands. Now, we show a slightly wider range of the spectra from 30 to -40 ppm.

Dear Reviewer #2

*******Reviewer #2 comment starts*******

Reviewer #2 (Remarks to the Author):

Review of “In situ tunable links between key Li-Si structural transformations and Coulombic reversibility” by K. Ogata et al.

In this study, the authors report an extensive investigation into the links between depth of discharge, Coulombic efficiency, and the formation of the Li₁₅Si₄ phase during cycling of silicon anodes for Li-ion batteries. A significant electrochemical effort, complemented by ex situ TEM, XRD, NMR, XAFS, and DFT, is pursued. The research shows that there is indeed a link between CE and the formation of the Li₁₅Si₄ phase (among other conclusions), which may be important for designing electrodes and full cells for application of Si in commercial cells.

This study presents an enormous amount of data, and the work is generally well done and important for progress of Si anodes. However, I expect that this paper will be of most interest to experts in the field of alloying anodes for Li-ion batteries, and of less interest to general materials/electrochemical scientists, as required by Nature Communications. The highly specialized nature of the manuscript is exemplified by the considerable use of acronyms, shortened sample names, and abbreviated labels which makes the manuscript difficult to follow in some points, even for experts. I understand the authors’ dilemma – the data are extensive and complex, and they have done a commendable job of organizing and presenting it. However, the very nature of the findings in this paper perhaps point to a more specialized and non-“communications” journal in which the findings can be presented in full. Based on these comments, I think it should be up to the editors’ discretion to determine whether this paper is suitable for Nature Communications.

A few additional suggestions:

-A minor note: in the introduction, the authors state that previous work has shown “large asymmetric VE owing to different Li-diffusivity through different c-Si facets.” While there was some initial confusion regarding this phenomenon, the consensus of the field is that this happens due to anisotropic reaction rates at different facets, not anisotropic diffusivity within the crystal. This should be corrected in the introduction.

-In fig. 2a, why does the DOD100% data have error bars, but the other data do not?

-The authors state that that the Li_{3.75}Si phase is clearly indexed in Fig. 4, but the SAED patterns are not labeled, and they are very small and difficult to discern.

-In Fig. 4g, the colored points are not labeled with 100%, 90%, 80% DOD.

-I appreciate the effort in Fig. 7 to simplify the conclusions with a schematic. However, this schematic is quite complex – there are no a, b, c labeled sections that are referred to in the caption, and it is difficult to know where to even start examining this figure. Also, are the colors in the pie charts labeled anywhere?

*******Reviewer #2 comment ends*******

Thank you very much for sparing your precious time amongst your busy schedule for reviewing our manuscript. We are glad to hear that you interpret and appreciate our key findings. Exactly as you suggest, the first manuscript was too lengthy and complex to concisely deliver our message to readers. We notice that these attempts can seriously distract readers’ understandings on our main points. Therefore, the manuscript is now much simplified with much less uses of abbreviations as shown in page 2. Also, so as to more clearly highlight our experimental scheme, new electrochemical findings, and new mechanistic findings, we also added **Supplementary Table 3–5** in *Supplementary information* (also shown in page 13-15 in this letter). Please also refer to our communications with Reviewer #1 as shown in **Response #1-3 (> Reviewer #1)** for a better understanding of our manuscript.

Response #4 (>Reviewer #2)

Reviewer #2 mentions that “This study presents an enormous amount of data, and the work is generally well done and important for progress of Si anodes. However, I expect that this paper will be of most interest to experts in the field of alloying anodes for Li-ion batteries, and of less interest to general materials/electrochemical scientists, as required by Nature Communications.”

We admit that the writing manner in the 1st draft was too complicated to concisely deliver our message to broader readers. Hence, in the modified draft (see page 2 for a modification list), we simplified the main text so that our key message can be delivered in more concise way. Actually, our message is consolidated into,

“ $c\text{-Li}_{3.75(+\delta)}\text{Si}$ is not just a bad guy, but can be a good person too”.

To prove this, we “**quantitatively and qualitatively revealed key correlations between CE evolution and the different Li-Si processes**” with the enormous dataset. Importantly, our claim is very contrast to previous works that typically conclude $c\text{-Li}_{3.75(+\delta)}\text{Si}$ has parasitic nature to degrade capacities and avoiding the formation improves the retention. As far as we notice, this is the first time that a positive aspect of $c\text{-Li}_{3.75(+\delta)}\text{Si}$ is highlighted.

Among numerous issues in Si anodes, Coulombic efficiency (CE) is actually one of the biggest issues remained to hinder the commercialisation. This is because, in reality, the volume expansion and corresponding capacity decay are somehow manageable at the state-of-the-art materials in pragmatic full cell designs. However, holding CE is NOT when Si volumetric concentration in the electrodes gets higher. An intuitive attempt to achieve higher CE is to engineer protective shells or a buffer medium around Si/Si-C composites to protect electrolyte invasion and consequent irreversible Li consumption. Nevertheless, electrolyte can unfortunately invade due to Li-ions’ transport nature (a necessity to be coupled with organic components) and/or due to gradual structural deformation even with internal-pore-engineered designs over iterative volume changes. We have been extremely sensitive to these issues at both research and development levels. Hence, scenarios that Si can be anyway exposed to electrolytes need to be considered for understanding CE fundamentals. Despite the importance of CE, the fundamental works are extremely scarce. We reckon that such scarcity may partly originate from the difficulty (particularly in academic research environments) of satisfying **Prerequisite #1–2**“ in **Response #1 (>Reviewer #1)**; you may notice that even a minor temperature fluctuation by air-conditioner in a cyclers’ room can fluctuate electrochemical profiles. Or, the profiles fluctuate by interference of entangled electric wires.

Moreover, the findings in this study can rather directly connect to new cell-design strategies, trading-off the CE boost and the sacrificial capacity loss by e.g. N/P controls and prelithiation, for advanced Li-ion batteries as well as all-solid-state, Li-sulfur, and Li-metal systems (we can provide preliminary results if necessary).

In this sense, we believe that our findings on the CE fundamentals can attract more generic interests in this journal rather than being published in more specialised journals.

Response #5 (>Reviewer #2)

Reviewer #2 mentions "However, the very nature of the findings in this paper perhaps point to a more specialized and non-“communications” journal in which the findings can be presented in full....."

As far as we notice, an article in Nature communications, despite the name "communications", is published in a full paper style with relevant text length to our modified manuscript.

Response #6 (>Reviewer #2)

A few additional suggestions:

-A minor note: in the introduction, the authors state that previous work has shown “large asymmetric VE owing to different Li-diffusivity through different c-Si facets.” While there was some initial confusion regarding this phenomenon, the consensus of the field is that this happens due to anisotropic reaction rates at different facets, not anisotropic diffusivity within the crystal. This should be corrected in the introduction.

That is exactly right and the description is our misinterpretation. Actually, as several recent *in situ* TEM findings (e.g. McDowell *et. al.*) show, the Li-Si process is reaction limited rather than diffusion limited. We now accordingly corrected the text.

-In fig. 2a, why does the DOD100% data have error bars, but the other data do not?

This is because the profiles for DOD70-90 % are based on lithiation capacities which are pre-programmed in the cyclor. For CE, all the error bars are within a range of $\sim\pm 0.1\%$, which is now clearly mentioned in the corresponding figure captions (**Figure 1** and **Supplementary Fig. S 4**), and also written in **Methods**.

-The authors state that that the $\text{Li}_{3.75}\text{Si}$ phase is clearly indexed in Fig. 4, but the SAED patterns are not labeled, and they are very small and difficult to discern.

This is now modified such that the label and SAED/SADP are clearly seen.

-In Fig. 4g, the colored points are not labeled with 100%, 90%, 80% DOD.

This is accordingly corrected.

-I appreciate the effort in Fig. 7 to simplify the conclusions with a schematic. However, this schematic is quite complex – there are no a, b, c labeled sections that are referred to in the caption, and it is difficult to know where to even start examining this figure. Also, are the colors in the pie charts labeled anywhere?

Thanks for understanding our efforts. This is a very useful suggestion. As you suggest, we segmented the schematics into three parts, (a-c), so that reader can follow the logistics in the schematics.

Reviewers' comments:

Reviewer #2 (Remarks to the Author):

I have reviewed the authors' response and the revised manuscript in detail.

As I stated in my original review, I do believe that the paper has interesting results, and the connections made between the extensive electrochemical data and the other characterization methods are compelling. And as the authors point out, understanding/controlling/improving CE is the critical aspect of getting Si anodes in batteries.

The revisions do not change my conclusion that this manuscript is of primary interest to researchers in the field of silicon anodes for batteries, and not a broad audience. In short: the paper is useful, but it is extremely dense and not accessible to a broad audience.

If Nature Communications wants to publish papers with interesting results for a specific sub-field, then I suggest to publish this paper.

If Nature Communications wants to publish only papers that might be of interest to scientists across multiple fields, then don't publish this paper.

Reviewer #3 (Remarks to the Author):

Reviewer Comment

I agree with both reviewers #1 and #2's assessment on the manuscript. More specifically, the authors present enormous amount of data in the MS, and there is no doubt that the data are of high quality. The authors must have put a lot of time and effort in this MS, and I do appreciate their effort. However, the paper is still hard to read because they put too many details in it. As such it is easy for readers to get lost of the big picture. There is just too many details and too much information, and some of the figures are so complicated that I could not understand what they stand for even after extensive effort of reading by myself (e.g. Fig. 7).

Overall I am in the opinion that the editor may give the authors another chance to further improve the readability of the MS by getting rid of the enormous details, distilling the main points, simplifying the figures, considering their great effort in the work.

#1. Point-by-point responses to reviewer's comments

Here we answer your concerns point-by-point as followings. Your comments are highlighted by red.

Reviewer #2 (Remarks to the Author):

I have reviewed the authors' response and the revised manuscript in detail.

As I stated in my original review, I do believe that the paper has interesting results, and the connections made between the extensive electrochemical data and the other characterization methods are compelling. And as the authors point out, understanding/controlling/improving CE is the critical aspect of getting Si anodes in batteries.

The revisions do not change my conclusion that this manuscript is of primary interest to researchers in the field of silicon anodes for batteries, and not a broad audience. In short: the paper is useful, but it is extremely dense and not accessible to a broad audience.

If Nature Communications wants to publish papers with interesting results for a specific sub-field, then I suggest to publish this paper.

If Nature Communications wants to publish only papers that might be of interest to scientists across multiple fields, then don't publish this paper.

Reviewer #3 (Remarks to the Author):

Reviewer Comment

I agree with both reviewers #1 and #2's assessment on the manuscript. More specifically, the authors present enormous amount of data in the MS, and there is no doubt that the data are of high quality. The authors must have put a lot of time and effort in this MS, and I do appreciate their effort. However, the paper is still hard to read because they put too many details in it. As such it is easy for readers to get lost of the big picture. There is just too many details and too much information, and some of the figures are so complicated that I could not understand what they stand for even after extensive effort of reading by myself (e.g. Fig. 7).

Overall I am in the opinion that the editor may give the authors another chance to further improve the readability of the MS by getting rid of the enormous details, distilling the main points, simplifying the figures, considering their great effort in the work.

We again appreciate your efforts on digesting our dataset despite some difficulty in readability and accessibility. We also appreciate your acknowledgements on importance and credibility of our scientific claims. In the following, we explain our thoughts regarding your concerns.

Firstly, **Reviewer #2** concerns that our findings might be of interest only for Si anode researchers and may not be suitable for a publication in the journal. For this, let us highlight some state-of-the-art and realistic backgrounds around Si anodes with our best knowledge both in terms of industry and academic viewpoints.

Si anodes, more specifically, *Si-rich* anodes, as shown in this study, are one of very few remaining options that can certainly leap energy density of the rechargeable cells. The density in cutting-edge commercial LiB cells currently lies around near ~ 700 Wh/L with either LCO (typically for consumable electronics) or NCA/NCM (e.g. for larger applications) coupled with Gr (a few manufacturers optionally blend Gr with prelithiated 3–5wt% SiO_x). However, as you may know well, a learning curve of the density with the current material strategies is nearly saturated or sluggish over the last 5–10 years and most likely to be capped around ~ 750 Wh/L any soon. With the use of Si-rich anodes (probably combined with state-of-the-art prelithiation technologies), the density can reach up to 900–1000 Wh/L or higher. Since, *mass* commercialization of Li-metal and Li-air batteries would still necessitate a longer time scale to fulfil present performance requirements and safety constrains, we believe that Si-rich anodes can be one of the major players in the market for the next decade(s). Importantly, Si-rich anodes can be coupled *not only* in next-gen LiB *but also* Li-sulphur batteries with solid-state/non-aqueous electrolytes.

However, as we pointed out in the last “Point-by-point responses to reviewers” as well as **Introduction** in the main text, one of the most critical issues for the anodes remains to be explored, which is the poor Coulombic efficiency (CE) of the complex Li–Si processes. Notably, sustaining capacity retention in Li-unlimited half-cells is rather handled by elaborated various Si-C composites, yet CE is NOT for longer-term cycles particularly when Si surface is gradually exposed to electrolytes. This becomes more critical when Si concentration in the anodes is higher. As one of the most basic, it is the key to quantify and qualify the subtle yet accumulatively significant CE alterations by different Li–Si processes over longer-term cycles. However, unfortunately, most of the recent works are not even designed to probe the alterations (explained in **Prerequisite #1–2** in the previous letter). Consequently, the insights are surprisingly scarce despite primary significance of the issue in the fields.

In this manuscript, we succeeded in providing unprecedented insights on this. We quantitatively and qualitatively resolved how the irreversible consumption is correlated with different Li–Si processes (separating *a–a* volume changes from the *a–c* phase transformations) revealing structural/electrochemical mechanism behind the correlation. We highlighted that the most efficient way to minimize the cumulative irreversible Li consumption can lie in repeating *c*- $\text{Li}_{3.75(+\delta)}\text{Si}$ formation/decomposition, which is typically recognized as one of a degradation factor in the anodes. Importantly, we for the first time uncovered a positive aspect of *c*- $\text{Li}_{3.75(+\delta)}\text{Si}$. Further, we briefly yet concretely proposed how we can incorporate these insights into real cell systems in the end of **Discussion** part, which can be expanded to the various batteries. Therefore, we believe that our findings will provide significant insights and practical benefits to the fields and accordingly attract broader readers if the presentation manner in the manuscript secures sufficient readability and accessibility.

Secondly, as **Reviewer #3** points out, we agree with the remaining complexity in the last manuscript that may have hindered readability and accessibility to attract broader readers. We also recognized that this is a serious issue. Hence, we had attempts to receive various feedbacks on our manuscript from researchers in the relevant fields, to extract bottlenecks of the issue, and accordingly modified the manuscript. We describe our modification strategy and attempts in page 4, and show details of the modification in page 5–8.